# Trait-associated noncoding variant regions affect *TBX3* regulation and cardiac conduction

Jan Hendrik van Weerd[1], Rajiv A Mohan[1,2], Karel van Duijvenboden[1], Ingeborg B Hooijkaas[1], Vincent Wakker[1], Bastiaan J Boukens[1,2], Phil Barnett[1], Vincent M Christoffels[1]*

[1]Department of Medical Biology, Amsterdam Cardiovascular Sciences, Amsterdam University Medical Centers, University of Amsterdam, Amsterdam, Netherlands; [2]Department of Clinical and Experimental Cardiology, Amsterdam Cardiovascular Sciences, Amsterdam University Medical Centers, University of Amsterdam, Amsterdam, Netherlands

**Abstract** Genome-wide association studies have implicated common genomic variants in the gene desert upstream of *TBX3* in cardiac conduction velocity. Whether these noncoding variants affect expression of *TBX3* or neighboring genes and how they affect cardiac conduction is not understood. Here, we use high-throughput STARR-seq to test the entire 1.3 Mb human and mouse *TBX3* locus, including two cardiac conduction-associated variant regions, for regulatory function. We identified multiple accessible and functional regulatory DNA elements that harbor variants affecting their activity. Both variant regions drove gene expression in the cardiac conduction tissue in transgenic reporter mice. Genomic deletion from the mouse genome of one of the regions caused increased cardiac expression of only *Tbx3,* PR interval shortening and increased QRS duration. Combined, our findings address the mechanistic link between trait-associated variants in the gene desert, *TBX3* regulation and cardiac conduction.

**\*For correspondence:**
v.m.christoffels@amsterdamumc.nl

**Competing interests:** The authors declare that no competing interests exist.

## Introduction

Genome-wide association studies (GWAS) increasingly implicate common variation in the human genome with traits and complex diseases, including traits reflecting cardiac conduction properties. The majority of trait-associated genetic variants (single nucleotide polymorphisms; SNPs) is found in noncoding genomic DNA, and is thought to affect the function of *cis*-regulatory elements (REs) controlling the activity of their target gene promoters (*Deplancke et al., 2016*; *Maurano et al., 2012*; *Schaub et al., 2012*). Expression quantitative trait locus (eQTL) analyses show that common variation can lead to both down- and upregulation of target gene expression (*Gutierrez and Chung, 2016*; *Hsu et al., 2018*; *Roselli et al., 2018*). Yet, the mechanisms underlying the effect of such variation on RE function and gene expression are poorly understood.

The atrioventricular conduction system (AVCS) conducts the electrical impulse from atria to ventricles and synchronizes ventricular activation, thus orchestrating the rhythm of the heart. Disrupted AVCS function can lead to life-threatening cardiac arrhythmias and hypertrophy (*Aeschbacher et al., 2018*; *Cheng et al., 2009*). Common SNPs in the gene desert upstream of *TBX3* have been strongly associated with both PR interval and QRS duration (*Pfeufer et al., 2010*; *Sotoodehnia et al., 2010*; *van der Harst et al., 2016*; *van Setten et al., 2018*; *Verweij et al., 2014*), ECG parameters directly reflective of AVCS function and representing the time between the first moment of activation of the atria and the ventricles, and ventricular activation, respectively. The T-box transcription factor 3 (Tbx3) is specifically expressed in the AVCS, including the AV node, AV bundle and proximal bundle

branches, and plays a critical role in its development and function (*Bakker et al., 2012*; *Bakker et al., 2008*; *Frank et al., 2011*; *Singh et al., 2012*). Presumably, the SNPs upstream of *TBX3* affect the expression of *TBX3* or other nearby genes, such as *TBX5*, which also plays a key role in AVCS development and function (*Arnolds et al., 2012*; *Burnicka-Turek et al., 2020*), or *MED13L*, encoding a subunit of the ubiquitously expressed Mediator complex involved in transcriptional activation (*Asadollahi et al., 2013*; *van Haelst et al., 2015*) and recently implicated in heart rate recovery after exercise (*Ramírez et al., 2018*; *Verweij et al., 2018*).

Efforts to elucidate the mechanisms underlying the transcriptional regulation of AVCS genes and the role of genetic variation are challenging, as the availability of relevant cells for functional and (epi)genomic experiments is hampered by the very small proportion of $TBX3^+$ AVCS cardiomyocytes in the heart and the heterogeneous cellular composition of the AVCS (*Aanhaanen et al., 2010*; *Hoogaars et al., 2004*). Along this line, eQTL data for *TBX3* in cardiac tissue is not available, and AVCS tissue from human hearts is not expected to become available soon. Moreover, suitable AVCS-like cell lines, derived from stem cells or immortalized primary cells do not exist. Current cardiac epigenomic datasets from humans are mainly derived from whole heart or heart compartment tissue and are not suitable for the identification of AVCS REs, as the proportion of AVCS cells in these tissues is extremely small.

In this study, we aimed at circumventing the unavailability of AVCS cells to elucidate how the PR interval- and QRS duration-associated variants in the gene desert in-between *TBX3* and *MED13L* affect the regulation of gene expression and AVCS function. Utilizing mouse AVCS-specific assay for transposase-accessible chromatin sequencing (ATAC-seq) and human and mouse *TBX3/TBX5* locus-wide self-transcribing active regulatory region sequencing (STARR-seq), we identified multiple regulatory regions within two risk loci upstream of *TBX3* that are activated by factors involved in AVCS development and drive gene expression in vitro. We assessed the presence of associated variants in these REs and tested their effect on RE activity. Using modified BACs containing the mouse orthologues of the respective risk loci, we show that each region independently drives reporter expression in the AVCS in transgenic mice. We deleted the mouse homologous regions of both human risk loci using CRISPR/Cas9-mediated genome editing and show that deletion of the proximal domain resulted in a two-fold increase in the expression of *Tbx3*, but not *Tbx5* or *Med13l*, specifically in the AVCS, and in decreased PR interval and increased QRS duration. Thus, we used a systematic approach to identify REs driving *TBX3* expression in the small number of cells that make up the AVCS, and provide evidence that a genomic noncoding region associated with PR interval is involved in the regulation of *Tbx3* expression and AVCS function in vivo.

## Results

### GWAS variation in gene desert upstream of TBX3

Common genomic variants associated with PR interval and QRS duration have been identified in the region of the human *TBX3/TBX5* gene cluster (*Pfeufer et al., 2010*; *Sotoodehnia et al., 2010*; *van der Harst et al., 2016*; *van Setten et al., 2018*). Using publicly available human Hi-C data, we defined the topologically associating domain (TAD) of this gene cluster. *TBX3* and its upstream gene desert are organized in a TAD of approximately 1.3 Mb that is delimited by binding sites for CCCTC-binding factor (CTCF; a zinc-finger protein associated with enhancer-promoter looping and TAD boundary formation [*de Wit et al., 2015*; *Nora et al., 2017*]) and physically separated from that of the neighboring gene *MED13L*, corresponding to the organization of the murine loci in cardiac embryonic tissue (*van Weerd et al., 2014*; *Figure 1A*). Sequences within the TADs of *TBX3* and *TBX5* physically interact with each other to some extent. To delineate the GWAS risk loci, we mapped the associated variants to the human reference genome. Variants associated with both PR interval (*van Setten et al., 2018*) and QRS duration (*van der Harst et al., 2016*) are located in a region approximately −261 kb to −221 kb upstream of *TBX3* (variant region (VR) 1). A second region (VR2) ranging from −85 kb to −6 kb upstream of *TBX3* harbors SNPs associated with only PR interval. SNPs associated with QRS duration were also found downstream of *TBX3*. The location of these variants within the *TBX3* TAD suggests that they may affect putative REs that regulate the expression of *TBX3* and possibly *TBX5* in the heart, causing affected conduction parameters.

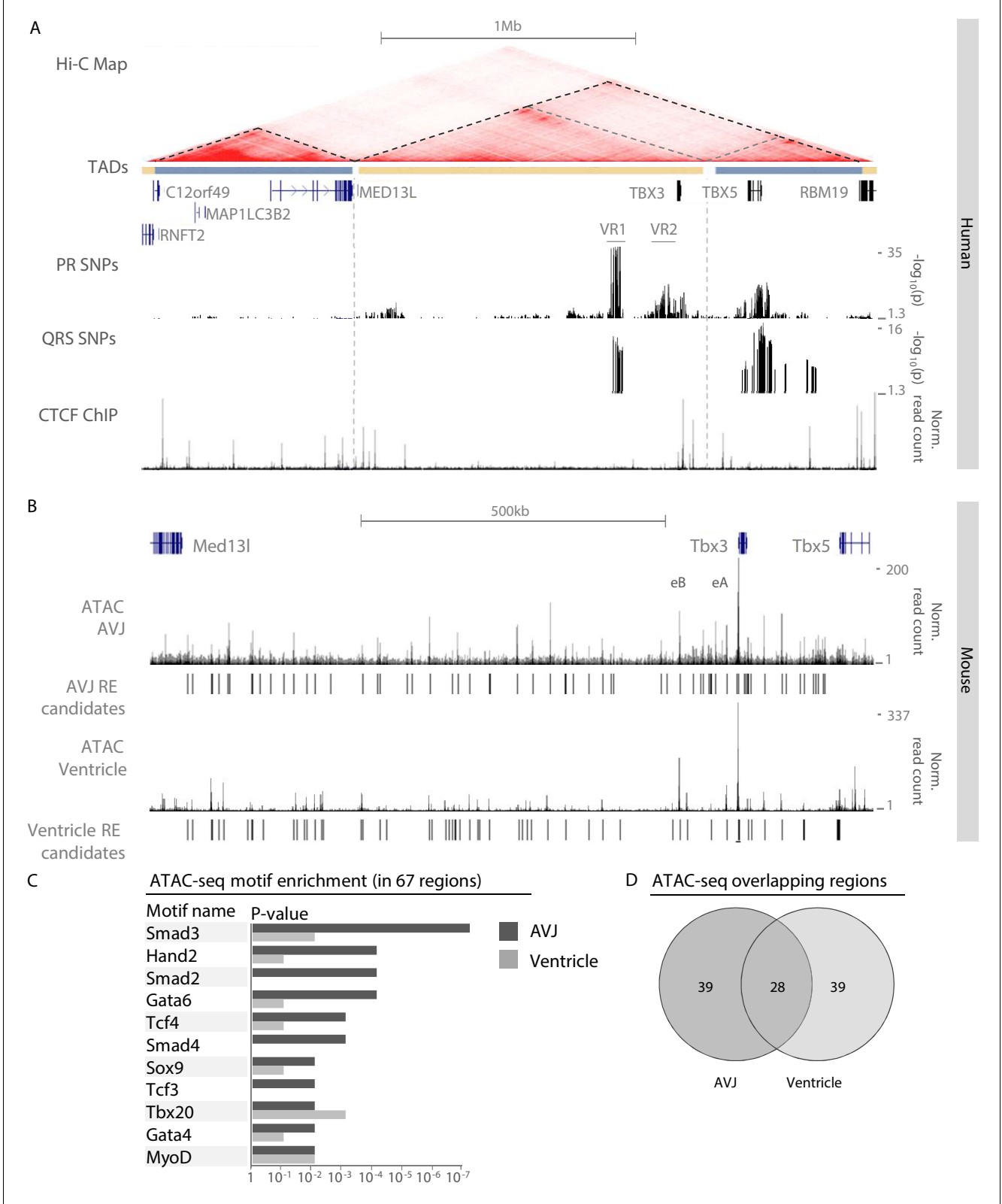

**Figure 1.** Identification of AVCS-specific accessible chromatin within the *Tbx3* locus harboring GWAS-associated variant regions. (**A**) UCSC genome browser view depicts the genomic location of human *TBX3* and neighboring genes. GWAS SNPs associated with PR interval (*van Setten et al., 2018*) and QRS duration (*van der Harst et al., 2016*) are plotted as –log10 of their p-values (lower cut-off 1.3 (p=0.05)) and depict two variant regions (VR1 and VR2) upstream of *TBX3*. CTCF ChIP-seq track shows binding sites for CTCF, corresponding to the boundaries of the topologically associating

*Figure 1 continued on next page*

*Figure 1 continued*

domains. The Hi-C map is derived from a human lymphoblastoid cell line (*Rao et al., 2014*) and reveals separated domains for *TBX3* and neighboring *MED13L* and *TBX5* (dark grey dashed lines). Within the *TBX3/TBX5* TAD, the *TBX3* and *TBX5* loci form sub-TADs (light grey dashed lines). (**B**) Genome browser view depicting embryonic Tbx3[+] AVCS-specific (ATAC_AVCS) and ventricular (ATAC Ventricle) accessible chromatin regions (*van Duijvenboden et al., 2019*) within the murine *Tbx3* locus. AVCS and Ventricle RE candidates depict selected ATAC regions. (**C**) HOMER motif enrichment analysis on 67 AVCS and Ventricle ATAC regions (depicted in **B**) reveals binding motifs for Smad, Gata and Tcf factors are enriched in AV junction accessible regions compared to ventricular accessible regions. (**D**) Overlap of 67 ATAC_AVJ regions within *Tbx3* locus with 67 ATAC-ventricle regions.

## Identification of accessible regulatory elements within the murine Tbx3 locus

To identify REs driving *Tbx3* expression in the AVCS that are possibly affected by genetic variation within VR1 and VR2, we performed ATAC-seq (assay for transposase-accessible chromatin, followed by sequencing) (*Buenrostro et al., 2013*) on *Tbx3*+ FACS-purified AVCS cardiomyocytes from murine E13.5 *Tbx3*^Venus/+ microdissected AV junctions. 67 genomic regions within the *Tbx3/Tbx5* regulatory domain, as determined by chromatin conformation capture (*van Weerd et al., 2014*), are accessible in AVCS cardiomyocytes, including the previously identified AV canal enhancers Tbx3-eA and Tbx3-eB (*van Weerd et al., 2014*; *Figure 1B*). To identify transcription factors potentially involved in the regulation of *Tbx3* in the AVCS, we performed a motif enrichment analysis on the 67 regions within the *Tbx3* locus and compared it to the analysis of an equal number of accessible regions in ventricular tissue within the locus (*van Duijvenboden et al., 2019*). Among the highest scoring binding motifs in embryonic AVCS cells are motifs for transcription factors important for cardiogenesis, including Hand2, Sox9 and Tbx20 (*Figure 1C*). Interestingly, AVCS-specific accessible sequences were enriched for motifs for members of the Smad and Gata families of transcription factors (Smad2-4, Gata4/6), involved in the activation of regulatory sequences (including Tbx3-eA) specifically in the AV canal (*Stefanovic et al., 2014*), and for Tcf factors (Tcf4, Tcf3) acting downstream of canonical Wnt-signaling to regulate AV canal specification (*Gillers et al., 2015*; *Figure 1C*, *Supplementary file 1*-Supplementary Table 1). Of the 67 AVCS regions, 28 overlapped with accessible ventricular regions (*Figure 1D*). Analysis of enriched motifs in the 39 AVCS- or ventricle-specific regions did not result in more pronounced tissue-specificity (not shown), prompting us to continue with the 67 AVCS regions for further analysis.

## Identification of active regulatory sequences in the mouse and human TBX3 locus

The lack of AVCS-specific epigenomic datasets from mouse or human tissue and the absence of readily available human AVCS-relevant cells hamper efforts to identify relevant REs. To overcome this deficit, we utilized our finding that AVCS-accessible regions within the *Tbx3* locus are enriched for binding sites for Smad/Gata and Tcf (Wnt) transcription factor binding, involved in AVCS gene regulation (*Gillers et al., 2015*; *Stefanovic et al., 2014*). Using bacterial artificial chromosomes (BACs) spanning the entire human (+ / - 1.3 Mb; 17 BACs) and murine (+ / - 1 Mb; 11 BACs) *TBX3* locus, we performed self-transcribing active regulatory region-sequencing (STARR-seq) (*Arnold et al., 2013*) to scan for regulatory potential locus-wide (*Figure 2A*). The sequenced input libraries revealed an evenly distributed coverage throughout the locus, indicating that the entire locus is sufficiently and to an equal extent represented in the libraries (DNA input; *Figure 2—figure supplement 1*). Regions where two BACs overlap show an approximate doubling of the number of sequencing reads.

As AVCS-like cells are not available for large-scale and high-efficiency transfections, we transfected our STARR-seq libraries in COS-7 cells (a monkey kidney-derived fibroblast cell line), allowing us to assess the response of sequences within the libraries with high transfection efficiency specifically to the co-transfected transcription factors. Co-transfection of the libraries with pcDNA3.1 as control (m/hSTARR_ctrl) revealed multiple active sequences, indicative of REs capable of driving reporter gene expression in COS-7 cells (*Figure 2—figure supplement 1*). To find Smad/Gata and Wnt-responsive REs, we co-transfected the murine and human STARR libraries with Smad/Gata factors (m/hSTARR_SG4) or with Tcf-factors (m/hSTARR_Wnt), respectively. Regions with a log2 fold change of >0.58 (fold change >1.5) of SG4/Wnt-co-transfected activity over pcDNA-co-transfected

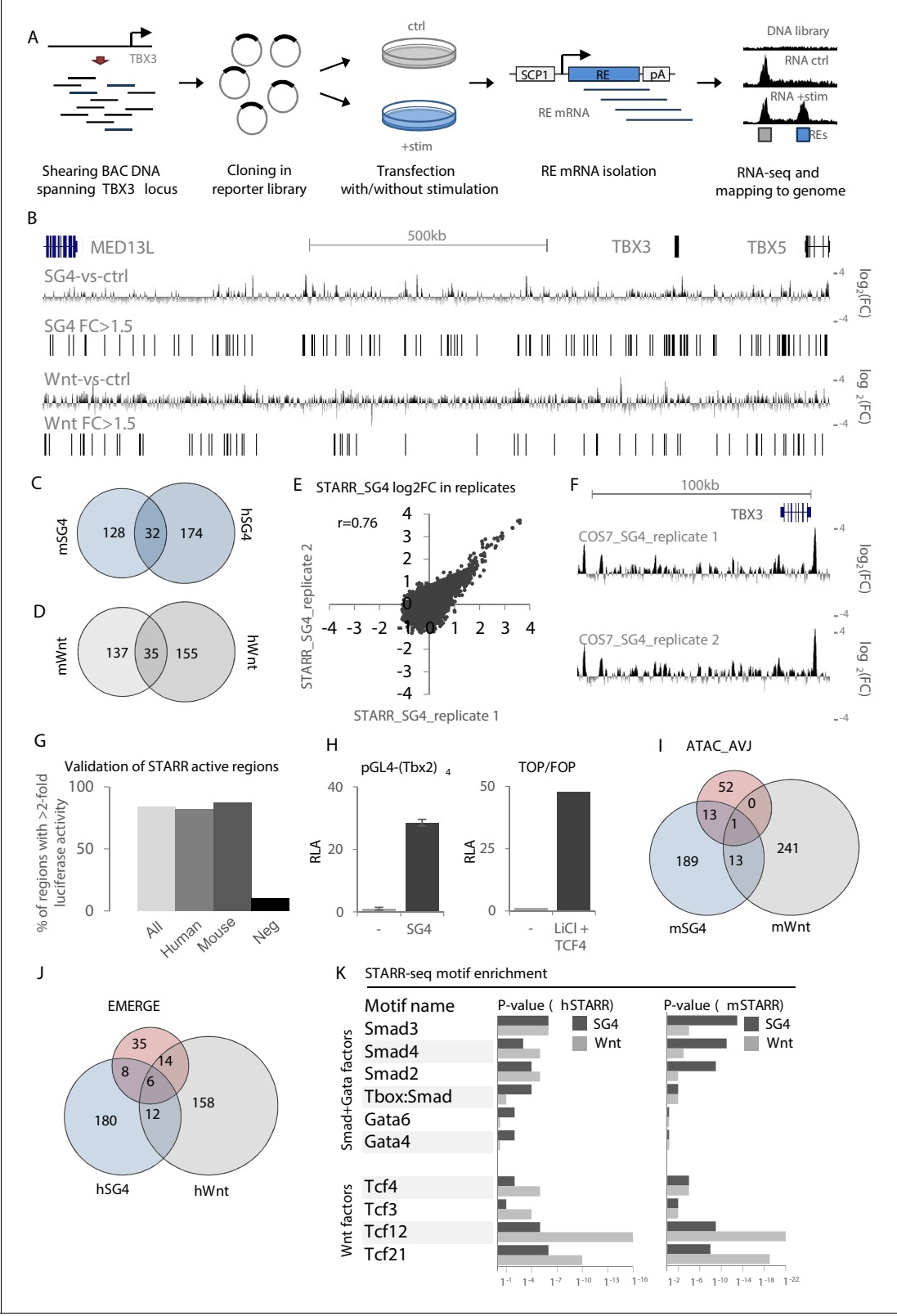

**Figure 2.** Validation and characterization of STARR-seq datasets. (**A**) Overview of the STARR-seq procedure on the murine/human *TBX3/TBX5* locus. BACs spanning the entire murine or human *TBX3/TBX5* locus are sheared to fragments of approximately 500–1000 bp. Fragments are recombined in the pSTARR expression vector to generate the STARR-seq libraries, which are transfected in cells with or without stimulation, that is with pcDNA (control) or expression vectors for Smad/Gata or Tcf factors. Two days after transfection, total RNA is isolated from all cells. mRNA transcripts are

*Figure 2 continued on next page*

*Figure 2 continued*

isolated from the pool of total RNA and next-generation sequenced on an Illumina MiSeq. Reads are mapped to the respective genomes. Comparison of stimulated over control data yields stimulus-responsive REs. (B) Overview of the human *TBX3/TBX5* locus with STARR-seq tracks for hSTARR_SG4 (SG4-vs-ctrl) and hSTARR_Wnt (Wnt-vs-ctrl). The displayed tracks represent log2 fold changes of enrichment of SG4/Wnt-stimulated regions over the unstimulated control. (C,D) Functional conservation of STARR-seq activity between murine and human fragments. Overlap of translated (mm9 to hg19) mSTARR_SG4 (160/216) (C) or mSTARR_Wnt (172/253) (D) regions with human hSTARR_SG4/Wnt regions. (E) Correlation between STARR-seq activities of replicate hSTARR_SG4 transfections in COS-7 cells. X- and y-axis depict log2 of the fold change of replicate 1 and replicate 2, respectively. (F) Genome browser views of hSTARR tracks for replicate transfections of libraries in COS-7 cells with co-transfection of SG4 factors show reproducibility of STARR-seq data. (G) Validation of selected active (log2FC >0.58 (FC >1.5)) hSTARR (n = 10), mSTARR (n = 9) and negative h/mSTARR (FC <1.5; n = 10) regions by luciferase assay. Y-axis represents percentage of tested fragments with activity in luciferase assay (FC >2 of reporter activity over empty vector). (H) Relative luciferase activity (RLA) of pGL4-(Tbx2)$_4$ (top) and the ratio of luciferase activity of TOP over FOP reporter (bottom) upon co-transfection with pcDNA ('-') and SG4- and Tcf-factors, respectively. Transfection in COS-7 cells; n = 4. (I,J) Overlap of mSTARR_SG4/Wnt regions with ATAC_AVCS regions (I) and of hSTARR_SG4/Wnt with hEMERGE regions (J). (K) HOMER motif enrichment analysis on human and murine STARR_SG4 and STARR_Wnt active (log2FC >0/58 (FC >1.5)) regions. Binding motifs for Smad/Gata factors are generally more enriched in STARR_SG4 regions, whereas STARR_Wnt regions are more enriched for motifs for Tcf factors in both human and murine datasets.

The online version of this article includes the following figure supplement(s) for figure 2:

**Figure supplement 1.** Overview of STARR-seq results in the human *TBX3/TBX5* locus.
**Figure supplement 2.** Discrepancy between STARR-seq activity and luciferase reporter activity of selected regions.

activity were considered to activate transcription. Combined, we found 216 SG4- and 257 Wnt-responsive regions in the murine locus, and 206 SG4- and 190 Wnt-responsive regions in the human locus (*Figure 2B*, *Figure 2—figure supplement 1*). The transcriptional potential of approximately 20% of the active mSTARR regions for both SG4- and Wnt-stimulation was functionally conserved between mouse and human, as the human homologues of these regions were active in the respective human STARR-seq libraries as well (*Figure 2C,D*). Duplicate transfections showed a strong correlation (*Figure 2E,F*). We performed transient transfection in COS-7 followed by luciferase assays with a subset of the identified regions to validate their regulatory potential. 84% of m/hSTARR regions (n = 19) drove luciferase activity >2 fold relative to the vector containing only the core promoter, compared to 10% of randomly selected negative regions (n = 10; m/hSTARR fold change <1.5; p=0.0007 (two-sample t-test)) (*Figure 2G*). These data confirm the positive signals (>1.5 fold over control) in STARR-seq represent fragments with enhancer activity. Among the active regions are both human and murine homologues of Tbx3-eA and Tbx3-eB. pGL4-(Tbx2)$_4$ (tandem repeat of a 380 bp Tbx2 enhancer responding to Smad- and Gata4 factors) (*Stefanovic et al., 2014*) and TOP/FOP (Wnt-signaling responsive TCF/LEF reporter assay) reporters showed strong luciferase activity upon stimulation with SG4- and Tcf-factors, respectively, validating that these factors activate SG4- and Tcf-responsive sequences upon transfection (*Figure 2H*). A subset of the active regions from mSTARR_SG4 and from mSTARR_Wnt overlapped both each other and ATAC_AVCS regions (*Figure 2I*), indicative of genomic regions both capable of activating reporter gene expression and being accessible in AVCS tissue. Similarly, a subset of hSTARR_SG4 and hSTARR_Wnt regions overlapped both each other and human EMERGE regions (*Figure 2J*, *Supplementary file 1*-Supplementary Table 2B).

To further justify the legitimacy of the chosen >1.5 threshold for fold activity and to validate that the identified regions were indeed activated by either SG4- or Wnt-factors, we performed transcription factor motif enrichment analysis and found that the 216 mSTARR_SG4 sequences were enriched for binding motifs for Smad and Gata factors when compared to an equal number of randomly selected mSTARR_Wnt active sequences. Vice versa, binding motifs for Tcf factors are enriched in the 257 mSTARR_Wnt regions when compared to mSTARR_SG4 regions (*Figure 2K*, *Supplementary file 1*-Supplementary Table 2A). These results indicate that 1) the regions we identified in the different libraries are activated by the respective cotransfected factors and 2) the threshold for regulatory activity of >1.5 fold change used to filter and select these regions is justified. Combined, these results suggest that multiple regions within both the human and murine *TBX3* locus are accessible specifically in AVCS tissue and respond to Smad/Gata and Wnt-signaling to activate reporter gene expression, indicating they are potentially involved in *TBX3* regulation in the AVCS in vivo.

## Identification of variant REs within VR1 and VR2

For the identification of regulatory sequences throughout the human *TBX3* locus potentially involved in AVCS-specific gene expression, we used three approaches. First, we combined the human STARR-seq regions activated by either SG4- or Tcf-factors throughout the *TBX3* locus, resulting in

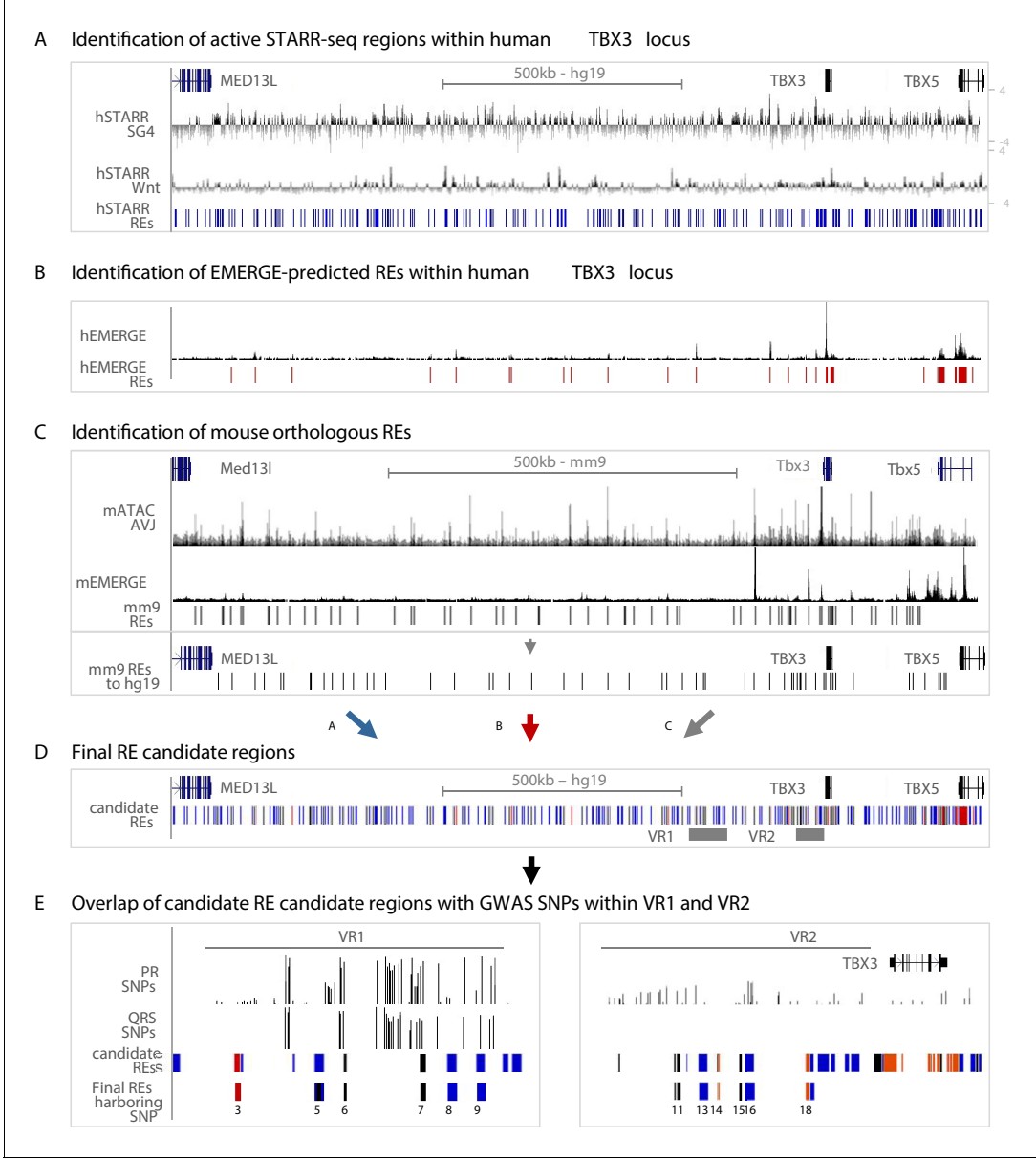

**Figure 3.** Approach for the identification of candidate REs potentially affected by GWAS variation. (**A**) Active STARR regions within the human *TBX3* locus, responding to either SG4- or Tcf-factors. Log2 ratios of the fold change of SG4- or Tcf-stimulated respons to pcDNA is depicted. Blue bars represent active regions (fold change >1.5) included for further analysis. (**B**) EMERGE enhancer prediction track (black) with selected enhancer candidates (red bars) included for further analysis. (**C**) Identification of candidate REs within the mouse *Tbx3* locus. Regions were selected based on ATAC_AVCS (top black track) and mouse EMERGE enhancer prediction (bottom black track). Selected regions (black bars) were translated to and plotted onto the human genome and included for further analysis. (**D**) Final RE candidate regions within the human *TBX3* based on the three criteria listed above. Blue bars: hSTARR regions; red bars: human EMERGE regions; black bars: mouse orthologous candidate regions translated to human genome. (**E**) Overlap of candidate regions from (**D**) within VR1 and VR2 with GWAS SNPs leads to final list of RE candidates potentially affected by GWAS variation.

The online version of this article includes the following figure supplement(s) for figure 3:

**Figure supplement 1.** STARR-seq of the murine *Tbx3/Tbx5* locus.

254 *h*STARR-REs (*Figure 3A*). Second, although epigenomic data from human AVCS tissue is scarce, we used cardiac-specific EMERGE enhancer prediction (*van Duijvenboden et al., 2016*) and found 24 putative cardiac-specific *h*EMERGE REs within the *TBX3* locus (*Figure 3B*). Third, we translated and mapped the genomic sequences of the 67 *m*ATAC_AVJ regions and of the cardiac-specific mouse EMERGE predicted enhancers to the human genome, resulting in 62 mouse orthologous regions (*Figure 3C*, *Figure 3—figure supplement 1A,B*). Combined, this approach resulted in 296 candidate RE regions in the human *TBX3* locus, based on sequence activity (hSTARR), enhancer prediction based on various cardiac-specific epigenomic datasets (hEMERGE), and conservation with active and AVCS-specific accessible genomic regions from the mouse genome (mouse orthologous regions) (*Figure 3D*). The previously identified human enhancer Tbx3-eA, driving AVCS expression and responding to Smad/Gata stimulation in vitro (*van Weerd et al., 2014*), is marked by all three criteria, supporting the validity of our approach (*Figure 3—figure supplement 1C*.

To assess whether these RE candidates harbor common variants associated with PR interval or QRS duration, we filtered for genomic location within VR1 or VR2 and found seven candidate variant REs within human VR1 and 13 candidate variant REs within VR2 (*Figure 3E*, *Figure 3—figure supplement 1*, *Supplementary file 1*-Supplementary Table 5). To assess the regulatory potential of the candidate REs and to elucidate whether their activity is affected by genomic variation, we overlapped these REs with PR interval- or QRS duration-associated variants (p>0.05) (*van der Harst et al., 2016*; *van Setten et al., 2018*), or variants in linkage disequilibrium (LD) with the most significantly associated variants within both regions ($r^2$ >0.5; *Figure 3E*, *Supplementary file 1*-Supplementary Table 4). Combined, we found six candidate REs in VR1 and 6 candidate REs in VR2 harboring one or multiple variants (*Figure 3E*).

We measured luciferase reporter activity of the major and minor haplotype of these RE candidates and found that all RE candidates except for *h*RE5 drove basal luciferase reporter activity in both HL-1 and COS-7 cells (n = 4; p<0.05); *h*RE5 decreased reporter activity (*Figure 4—figure supplement 1A*). Due to technical issues *h*RE13 was omitted from downstream experiments. We next measured regulatory activity of the risk and reference haplotypes for each RE. The risk allele of 4 REs (*h*RE5, *h*RE7, *h*RE8 and *h*RE18) increased basal reporter activity in HL-1 cells when compared to the reference allele, whereas the risk allele of *h*RE9, *h*RE11 and *h*RE15-16 decreased basal reporter activity in both HL-1 and COS-7 cells (*Figure 4A*). The risk allele of a subset of these regions also affected their SG4-/Wnt-induced activity (*Figure 4B*, *Figure 4—figure supplement 1B*). The SG4-dependent activity of *h*RE3 and *h*RE9 and the Wnt-dependent activity of *h*RE6 was increased by the risk allele, whereas the risk allele of *h*RE8, *h*RE15-16 and *h*RE18 decreased both SG4- and Wnt-dependent activity (*h*RE8, *h*RE18), or only Wnt-response (*h*RE15-16). Combined, these results show that multiple candidate REs within both VR1 and VR2 hold regulatory capacity in vitro, and that the activity of a subset of these regions (in both VR1 and VR2) is affected by the risk allele. We next analyzed whether potential TF binding motifs were disrupted or de novo created by the respective SNP in the variant region of the functionally affected REs to elucidate, thereby possibly explaining the differential effect of the risk allele on luciferase activity. None of the variants within the functionally affected fragments matched known TF binding motifs.

## Genomic regions upstream of TBX3 drive AVCS expression

The murine region homologous to human VR2 is located within a region that we previously demonstrated to harbor sequences driving *Tbx3* expression in the ventral, right-sided and dorsal (primordial AV node) aspect of the AV canal (BAC 366H17-GFP; −82 kb to +78 kb relative to *Tbx3*; *Figure 5*; *van Weerd et al., 2014*).Sequences within VR1 and VR2 are to a large extent evolutionary conserved between human and mouse, and VR2 harbors the AVC-specific RE (Tbx3-eA) that was functionally and structurally conserved between mouse and human (*van Weerd et al., 2014*; *Figure 5A,B*).To elucidate the involvement of the distal variant region (VR1) in the regulation of cardiac *Tbx3* expression, we set out to probe this domain for its regulatory potential by modifying a BAC spanning −245 kb to −75 kb upstream of *Tbx3* (459M16-GFP; *Figure 5B*). Although this BAC does not overlap VR2 and does not contain the previously identified RE (Tbx3-eA) driving AV node expression (*van Weerd et al., 2014*), immunohistochemistry on sections of *Tbx3*-459M16-GFP E14.5 embryos revealed GFP reporter expression throughout the entire AVCS in a pattern resembling that of endogenous *Tbx3*, including the full AV canal and prospective AV node, AV bundle and BBs (*Figure 5C*). Furthermore, expression was found in the eye, ear, genital tubercle, limbs, and

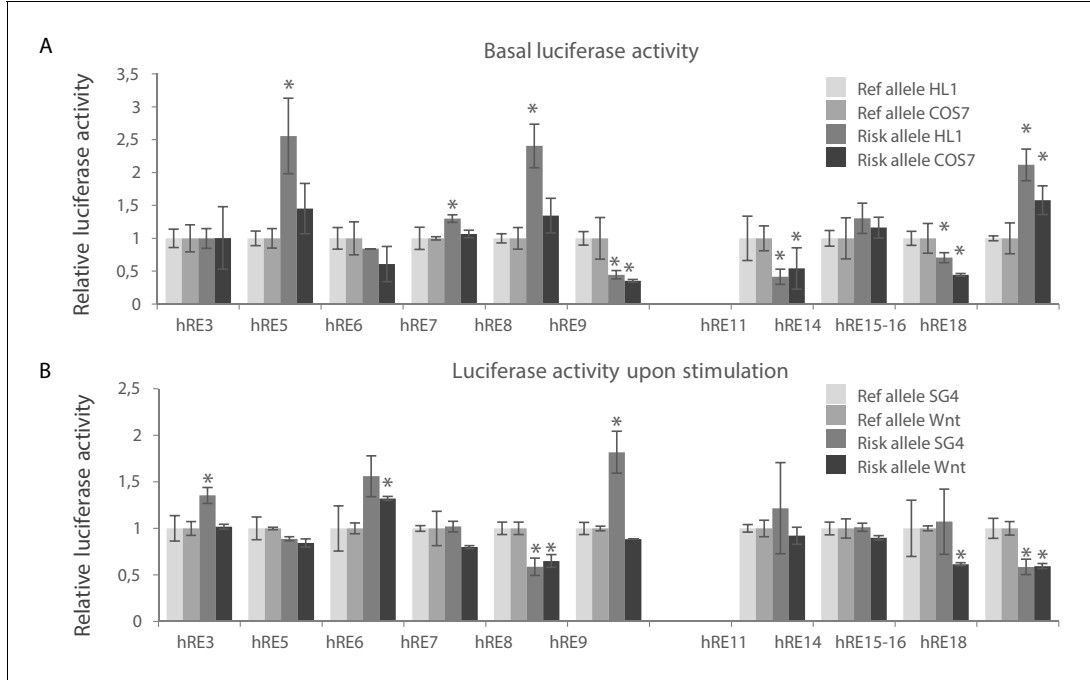

**Figure 4.** Identification of functional variants affecting RE candidate activity within human VR1 and VR2. (**A**) Basal luciferase activity of the reference and alternative alleles for each RE candidate in HL-1 and COS-7 cells. Luciferase activities were normalized to the activity of the reference allele for each fragment, in both HL-1 and COS-7 cells. (**B**) Relative luciferase activity of the reference and alternative alleles for each RE candidate upon stimulation with Smad- and Gata4 factors (SG4) and Tcf+LiCl factors (Wnt) in COS-7 cells. SG4/Wnt activity of the alternative alleles were normalized to the respective activity of the reference allele for each RE candidate. Transfections were performed in duplicates and replicated twice. Error bars represent standard deviations. *: p<0.05 (Student's t-test).

The online version of this article includes the following figure supplement(s) for figure 4:

**Figure supplement 1.** Regulatory activity of RE candidates.

mammary glands (*Figure 5—figure supplement 1*). This expression pattern is similar to that of previously described BAC 89K7-GFP, which only partially overlaps 459M16 and contains *Tbx3* and both the synergistically acting REs Tbx3-eA and Tbx3-eB (*van Weerd et al., 2014*; *Figure 5D*). These observations suggest that multiple physically separated regulatory regions independently drive AVCS expression (*Figure 5E*).

## Analysis of the function of VR1 and VR2 in vivo

To analyze the involvement of VR1 in endogenous regulation of gene expression and heart function, we used TALEN-mediated genome editing to delete the 69 kb homologous region (−245 kb to −176 kb relative to *Tbx3*) from the murine genome ($Tbx3^{\Delta VR1}$; *Figure 6A*). We validated the deletion by PCR and Sanger sequencing (*Figure 6—figure supplement 1A,B*). $Tbx3^{+/+}$, $Tbx3^{\Delta VR1/+}$ and $Tbx3^{\Delta VR1/\Delta VR1}$ mice were born according to Mendelian ratios (*Figure 6B*). We measured expression levels in microdissected AV junctions (containing the AV node and AV bundle) of E13.5 $Tbx3^{+/+}$ and $Tbx3^{\Delta VR1/\Delta VR1}$ hearts and found that expression of *Tbx3* or neighboring *Med13l* and *Tbx5* was not affected (*Figure 6C*, not shown for *Med13l* and *Tbx5*). To elucidate whether deletion of VR1 affects AVCS function postnatally, we performed surface ECGs on adult $Tbx3^{+/+}$ and $Tbx3^{\Delta VR1/\Delta VR1}$ mice and found no difference in PR interval or QRS duration between genotypes ($Tbx3^{+/+}$ n = 10, $Tbx3^{\Delta VR1/+}$ n = 12, $Tbx3^{\Delta VR1/\Delta VR1}$ n = 9; *Figure 6—figure supplement 2A–C*). To exclude a possible rescue effect of the sympathetic nervous system on AVCS function in the absence of VR1, we measured conduction parameters by ex vivo ECG recordings on Langendorff perfused hearts and again found no difference between $Tbx3^{+/+}$ and $Tbx3^{\Delta VR1/\Delta VR1}$ hearts (*Figure 6—figure supplement 2D, E*).

To assess the endogenous role of VR2 in the regulation of gene expression and AVCS function, we used CRISPR/Cas9-mediated genome editing to delete the 51 kb homologous region (−54 kb to

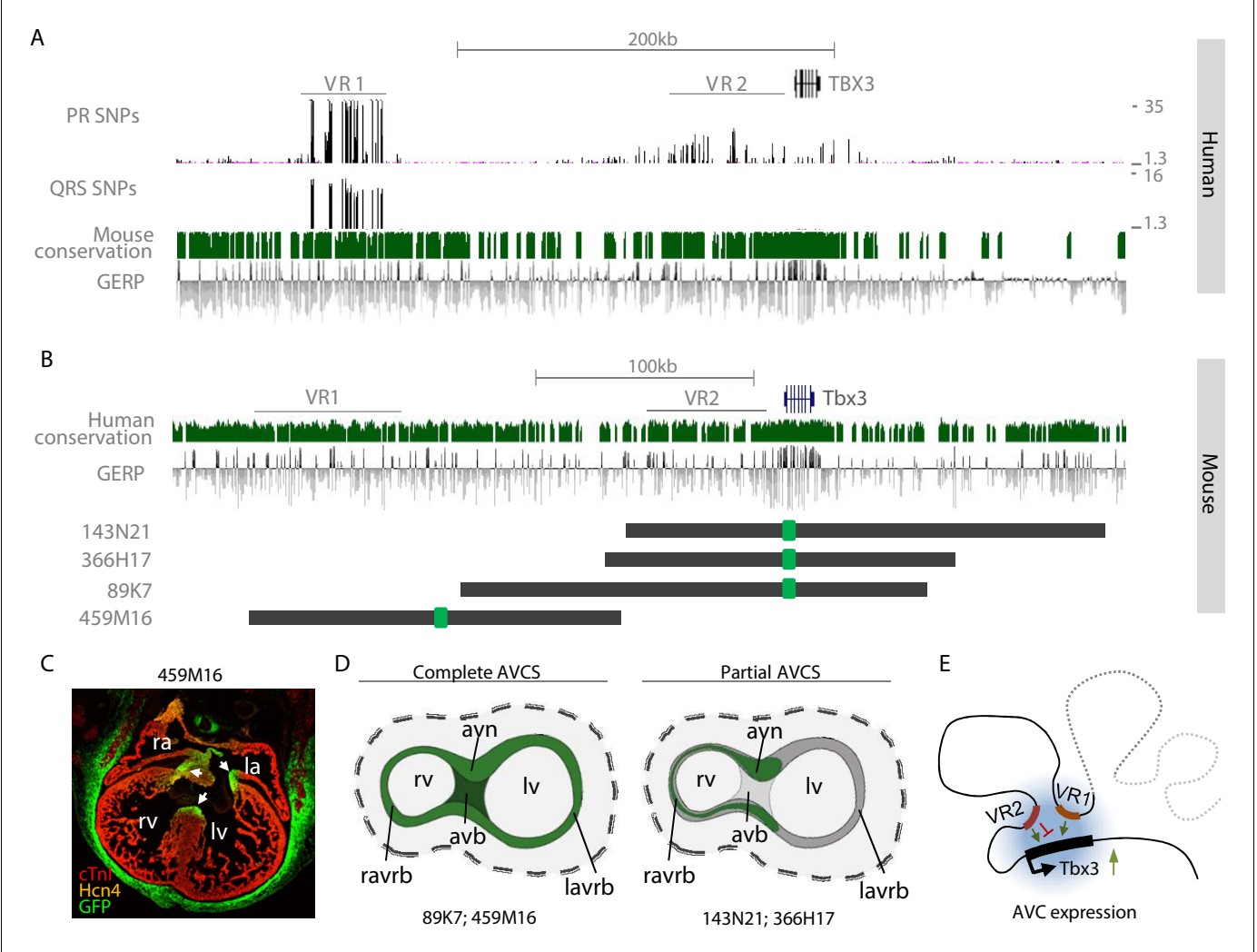

**Figure 5.** VR1 and VR2 are located within regulatory domains that redundantly drive AVCS expression. (**A**) Zoom-in of the human *TBX3* locus depicting distal variant region VR1, harboring SNPs for both PR interval and QRS duration, and proximal variant region VR2, harboring SNPs for PR interval. Mouse conservation (green) depicts genomic sequences conserved in the mouse genome; GERP depicts evolutionary constraint of genomic regions. (**B**) Homologous murine region within the *Tbx3* locus corresponding to the region depicted in (**A**). Homologous regions of VR1 and VR2 are depicted. Dark grey lines depict the locations of GFP-modified BACs, harboring either only VR2 (143N21, 366H17), both VR1 and VR2 (89K7) or only VR1 (459M16). (**C**) Immunohistochemistry on section through E14.5 459M16-GFP heart shows GFP reporter gene expression throughout the entire AVCS (white arrows). The Hcn4+ sinoatrial node does not express GFP. (**D**) Schematic overview of GFP-expression domains of BACs depicted in (**B**). The genomic region spanned by BACs 89K7 and 459M16, both harboring VR1, harbors regulatory elements driving expression throughout the complete AVCS including the atrioventricular bundle. The genomic region spanned by BACs 143N21 and 366H17 drive partial AVCS expression, not including left atrioventricular ring bundle and atrioventricular bundle expression. (**E**) Model of the topology of the *Tbx3* locus, depicting VR1 and VR2 both involved in regulation of *Tbx3* in the AV conduction system. Genomic sequences involved in the expression of *Tbx3* in the sinoatrial node are absent from these regions and likely reside more distally (grey dashed line). avb, atrioventricular bundle; avn, atrioventricular node; la, left atrium; lv, left ventricle; ra, right atrium; ravrb, right atrioventricular ring bundle; rv, right ventricle.

The online version of this article includes the following figure supplement(s) for figure 5:

**Figure supplement 1.** BAC 459M16-GFP recapitulates AVCS expression.

−3 kb relative to *Tbx3*) from the murine genome (*Tbx3*^ΔVR2). Again, we validated the deletion by PCR and Sanger sequencing (*Figure 6—figure supplement 1A,C*). At E13.5, the observed distribution of *Tbx3*^+/+, *Tbx3*^ΔVR2/+ and *Tbx3*^ΔVR2/ΔVR2 embryos followed Mendelian ratios (data not shown). Postnatally, however, the observed number of *Tbx3*^ΔVR2/ΔVR2 pups was approximately half of the expected number (p=0.004, χ2 test; *Figure 6D*). We analyzed expression levels of *Tbx3* in

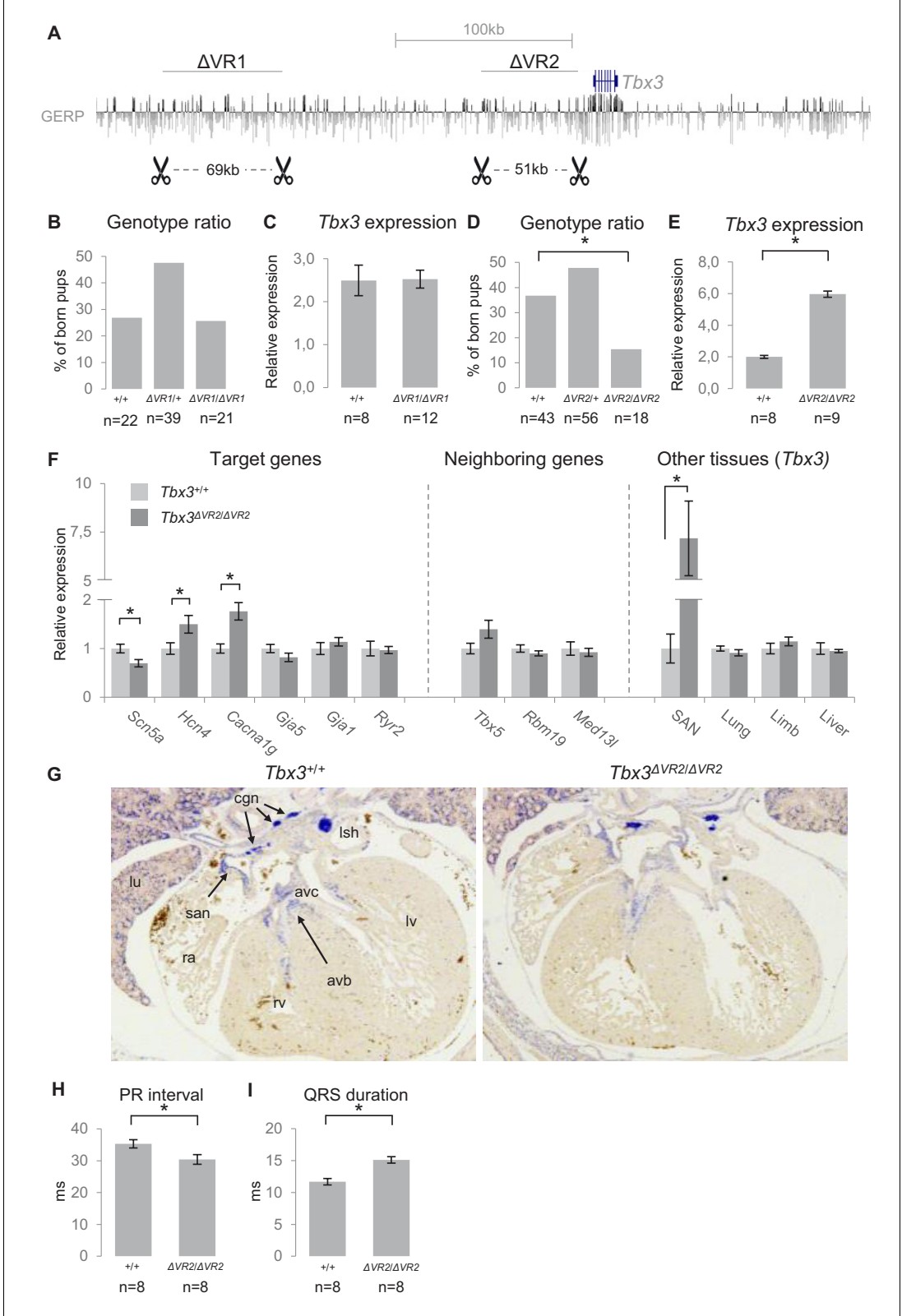

**Figure 6.** Genomic deletion of VR1 and VR2 in vivo. (**A**) Mouse genomic regions targeted by TALEN- (VR1) or CRISPR-Cas9- (VR2) mediated genome editing. (**B**) Genotype ratios of ΔVR1 mutant and wildtype born pups, depicted as percentage of total number of pups. (**C**) Relative expression levels of *Tbx3* in *Tbx3*[+/+] (n = 8) and *Tbx3*[ΔVR1/ΔVR1] (n = 12) microdissected E13.5 AV junctions. Expression levels are normalized to the expression of *Tbx2*. (**D**) Genotype ratios of ΔVR2 mutant and wildtype born pups, depicted as percentage of total number of pups, revealing the number of *Tbx3*[ΔVR2/ΔVR2] pups
*Figure 6 continued on next page*

Figure 6 continued

is lower than expected according to Mendelian ratios (*: p<0.01). (E) Relative expression levels of *Tbx3* in *Tbx3*[+/+] (n = 8) and *Tbx3*[ΔVR2/ΔVR2] (n = 9) microdissected E13.5 AV junctions, revealing increased expression of *Tbx3* in *Tbx3*[ΔVR2/ΔVR2] mutants (p<0.05). Expression levels are normalized to the expression of *Tbx2*. (F) Expression levels of target genes of *Tbx3* in microdissected E13.5 AV junctions, expression levels of neighboring genes in microdissected E13.5 AV junctions, and expression levels of *Tbx3* in other tissues in which Tbx3 is expressed, in *Tbx3*[+/+] (n = 8) and *Tbx3*[ΔVR2/ΔVR2] (n = 9) E13.5 embryos. Expression levels were normalized to the expression of *Tbx2* (AV junctions), *Isl1* (SAN) or *Hprt* (lung, limb, liver). *:p<0.05. (G) In-situ hybridization on sections of *Tbx3*[+/+] and *Tbx3*[ΔVR2/ΔVR2] E13.5 hearts shows expression of Tbx3 in the cardiac conduction system, including the sinoatrial node, atrioventricular canal, atrioventricular bundle and bundle branches. The expression pattern is not affected in *Tbx3*[ΔVR2/ΔVR2] hearts. (H,I) PR interval (H) and QRS duration (I) in *Tbx3*[+/+] (WT; n = 8) and *Tbx3*[ΔVR2/ΔVR2] (hom; n = 8) mice measured by surface ECG on male mice (2–4 months old). *:p<0.05. avb, atrioventricular bundle; avc, atrioventricular canal; cgn, cardiac ganglia; la, left atrium; lsh, left sinus horn; lu, lung; lv, left ventricle; ra, right atrium; rv, right ventricle; san, sinoatrial node.

The online version of this article includes the following figure supplement(s) for figure 6:

**Figure supplement 1.** Validation of genomic deletion of VR1 and VR2.
**Figure supplement 2.** Characterization of *Tbx3*[ΔVR1/ΔVR1] and *Tbx3*[ΔVR2/ΔVR2] mice.

microdissected AV junctions of E13.5 hearts and found an approximately 2.5-fold increase in expression in *Tbx3*[ΔVR2/ΔVR2] hearts compared to *Tbx3*[+/+] littermates (p=2*10$^{-11}$; n = 8 and n = 9, respectively; *Figure 6E*), respectively. *Tbx3*[ΔVR2/ΔVR2] hearts showed increased expression of *Hcn4* (p=0.04) and *Cacna1g* (p=0.004), ion channel-encoding genes involved in AVCS function, and decreased expression of *Scn5a* (p=0.02), the cardiac sodium channel Nav1.5-encoding gene involved in conduction, compared to hearts of wildtype littermates (*Figure 6F*). Expression levels of *Gja5* and *Gja1*, target genes of *Tbx3* in the AVCS and encoding the fast conducting gap junction proteins Cx40 and Cx43, respectively, were not affected (*Figure 6F*). To assess whether deletion of VR2 affects the expression of genes flanking *Tbx3*, we measured expression levels of *Tbx5*, *Rbm19* and *Med13l* and found that expression of these genes was unchanged (*Figure 6F*), suggesting VR2 only regulates *Tbx3* in the AVCS. *Tbx3* is also involved in the development of other tissues including the limbs, liver and lungs (*Davenport et al., 2003*; *Lüdtke et al., 2009*; *Lüdtke et al., 2013*). As the observed number of *Tbx3*[ΔVR2/ΔVR2] mice after birth was lower than expected, we asked whether *Tbx3* expression in these tissues is affected upon deletion of VR2, thereby potentially causing lethality. We measured *Tbx3* expression levels in sinoatrial nodes (SANs), forelimbs, liver and lungs of E13.5 embryos, and found increased expression only in the SAN (p=0.013). Expression in other tissues was not affected (*Figure 6F*). We next asked whether the upregulation of *Tbx3* in the embryonic AVCS is caused by increased expression in cells within its expression domain or by expansion of its expression domain in the heart. We performed in situ hybridization on E13.5 *Tbx3*[+/+] and *Tbx3*[ΔVR2/ΔVR2] hearts and observed no difference in expression pattern for *Tbx3* between *Tbx3*[+/+] and *Tbx3*[ΔVR2/ΔVR2] hearts (*Figure 6G*).

To assess the in vivo effects of deletion of VR2 on PR interval and QRS duration, we measured surface ECGs in *Tbx3*[+/+] and *Tbx3*[ΔVR2/ΔVR2] mice (n = 8 and n = 8, respectively). *Tbx3*[ΔVR2/ΔVR2] mice showed shortened PR intervals compared to *Tbx3*[+/+] mice (30.4 ms and 35.3 ms, respectively; p=0.025; *Figure 6H*, *Figure 6—figure supplement 2F*). The QRS duration in *Tbx3*[ΔVR2/ΔVR2] mice was increased compared to wildtypes (15.1 ms and 11.7 ms, respectively; p=0.004; *Figure 6I*, *Figure 6—figure supplement 2F*). Recordings of ex vivo ECGs on these hearts under Langendorff perfusion revealed a similar effect on PR interval and QRS duration, suggesting the autonomic nervous system is not involved in changed parameters upon deletion of VR2 (*Figure 6—figure supplement 2G,H*).

Taken together, these results show that deletion of VR1 in mice does not affect expression levels or AVCS function. Deletion of VR2 affects *Tbx3* expression in the AVCS and causes misregulation of target genes and affected conduction parameters.

## Discussion

Deciphering the mechanisms underlying gene regulation and the effects of trait-associated variation on these mechanisms is challenging for tissue types like the AVCS, which is small, heterogeneous and poorly available, and for which relevant model cells for functional and (epi)genomic experiments do not yet exist. Here, we functionally characterized two variant regions in the gene desert upstream

of *TBX3* that have been associated with PR interval and QRS duration, parameters for cardiac conduction, in the human population. Both regions are located within genomic domains that recapitulate AVCS expression in vivo. They harbor multiple active regulatory elements as determined by STARR-seq and analysis of epigenetic data, the activity of a subset of which was found to be affected by the risk haplotype. Deletion of the orthologue of the distal region from the mouse genome did not affect *Tbx3* expression or AVCS function. Deletion of the proximal region increased *Tbx3* expression in the AVCS and affected both PR interval and QRS duration in vivo.

A large number of PR interval and QRS duration-associated variants has been identified within loci close to genes important for cardiac conduction system development and function (*Pfeufer et al., 2010*; *Sotoodehnia et al., 2010*; *van der Harst et al., 2016*; *van Setten et al., 2018*; *Verweij et al., 2014*). Assigning function to these associations has been proven challenging. The vast majority of GWAS variation is located in non-coding genomic regions and therefore potentially affects the function of *cis*-REs, substantially impacting gene regulation and contributing to disease susceptibility (*Maurano et al., 2012*; *Schaub et al., 2012*). The identification of REs driving expression of genes active in the AVCS is hampered by the fact that the AVCS comprises only a small proportion of the heart. Consequently, the isolation of AVCS tissue for experimental procedures is challenging. eQTL analysis of associated variants is limited by the low number of available genotyped human AVCS samples, and AVCS-like cell lines for downstream functional testing of regulatory mechanisms are lacking. Using genome-wide murine AVCS-specific ATAC-seq and murine and human locus-wide functional enhancer screening using STARR-seq, we aimed to circumvent these issues and generate a means to identify REs potentially affected by GWAS variation.

GWAS typically identifies common variants, relatively frequently present in the human genome, and their effect size on the respective traits is usually small (*Manolio et al., 2009*; *McCarthy et al., 2008*). Haploinsufficiency of *TBX3*, *TBX5* and other transcription factor-encoding genes causes congenital defects (*Bamshad et al., 1997*; *Basson et al., 1999*; *Seidman and Seidman, 2002*) and is therefore not well tolerated, implying that even the modest effects of common variants on expression of such genes may have a relatively large effect on phenotypes. Furthermore, genes encoding developmental transcription factors are frequently under transcriptional control of multiple redundant REs (*Cannavò et al., 2016*; *Moorthy et al., 2017*; *Osterwalder et al., 2018*). The cardiac and limb expression of *Tbx3* involves multiple redundant or synergistically acting REs, which was previously shown by modified BAC and reporter transgenics (*Osterwalder et al., 2018*; *van Weerd et al., 2014*). Our current findings further substantiate regulatory redundancy in vivo. Two different BACs harbor the regulatory information for reporter gene expression in the AVCS and other tissues expressing *Tbx3*, whereas deletion of VR1 did not affect gene expression or AVCS function (*Osterwalder et al., 2018*; *van Weerd et al., 2014*). Combining this RE redundancy with the small effect size of associated variants, efforts to pinpoint individual causal variants in vivo or in modified (human) pluripotent stem cell-derived cardiac cell types will be challenging and with the currently available tools probably futile. Indeed, associations for many disease risk loci arise from multiple variants in LD that affect clusters of enhancers targeting the same gene (*Corradin and Scacheri, 2014*). Therefore, we set out to assess the effect of deletion of the entire variant regions within the *TBX3* locus rather than assessing the effect of individual SNPs, and found that the two *TBX3* risk loci harbor multiple active REs whose activity is affected by their respective risk alleles. Common genomic variation within these regions therefore probably affects the function of multiple redundant REs, rather than individual elements, that together orchestrate the complex regulation of *Tbx3* expression in the AVCS.

Despite the presence of four active REs within VR1, the activity of which is affected by PR and QRS duration associated variation, deletion of the VR1 orthologue from the murine genome did not lead to affected gene expression or AVCS function. Disease-causing non-coding variants in the human genome can affect RE activity and thereby gene expression by altering transcription factor binding (*Deplancke et al., 2016*; *Maurano et al., 2012*; *Schaub et al., 2012*), resulting in GWAS association. Genomic deletion of the entire VR1 removes the potentially affected REs from the genome and disrupts the topology of the locus, which could be functionally buffered by redundancy within the locus (*Cannavò et al., 2016*; *Long et al., 2016*; *Osterwalder et al., 2018*). Such redundancy is possibly not deployed in the case of only a variable nucleotide in an otherwise unaffected regulatory sequence (or genomic topology), providing a possible explanation for the absence of

effect we observed in $Tbx3^{\Delta VR1/\Delta VR1}$ mice (*Cannavò et al., 2016*; *Long et al., 2016*; *Osterwalder et al., 2018*).

Deletion of VR2 from the murine genome results in increased expression of *Tbx3*, and affected AVCS function. Regulation of the precise spatiotemporal activation of gene expression is orchestrated by various types of regulatory DNA elements including promoters, enhancers, silencers, insulators and other types of elements involved in conformation (*Heintzman et al., 2009*; *Long et al., 2016*; *Maston et al., 2006*). As we have previously identified an RE (Tbx3-eA) that activates gene expression in the AVCS (*van Weerd et al., 2014*) that is located within VR2 (*van Weerd et al., 2014*), the increased expression of *Tbx3* observed in $Tbx3^{\Delta VR2/\Delta VR2}$ hearts could be caused by a disruption of the fine balance between activating and repressive REs within VR2 that orchestrate the modulation of *Tbx3* regulation. Indeed, in the adult heart multiple regions within VR2 are marked by H3K27me3 (data not shown), a histone mark associated with transcriptional repression (*Gilsbach et al., 2014*; *Mozzetta et al., 2015*), suggesting VR2 potentially harbors multiple silencer elements.

The role of Tbx3 in conduction system function and development has been well demonstrated (*Bakker et al., 2008*; *Frank et al., 2011*; *Hoogaars et al., 2007*; *Horsthuis et al., 2009*). Tbx3 has a large number of target genes enriched for ion handling protein-encoding genes (*Bakker et al., 2012*; *Frank et al., 2011*; *Horsthuis et al., 2009*). As such, hundreds of genes that potentially influence the electrophysiological properties of the AVCS are likely to be deregulated in $Tbx3^{\Delta VR2/\Delta VR2}$. Not only *Tbx3* insufficiency (*Bakker et al., 2008*; *Frank et al., 2011*; *Horsthuis et al., 2009*) but also over-expression (this study, *Burnicka-Turek et al., 2020*) can be detrimental, suggesting proper AVCS function depends on strictly balanced Tbx3 levels. The balance between Tbx3 (imposing a nodal gene program) and Tbx5 (imposing a fast-conductive gene program) was found to be important in the function and gene regulation of the VCS (i.e. AV bundle and branches) (*Burnicka-Turek et al., 2020*). This balance model would predict conduction slowing when Tbx3 levels are increased. However, we observed shortened PR intervals indicative of faster AVCS conduction in $Tbx3^{\Delta VR2/\Delta VR2}$ mice (which have 2–3 fold increased *Tbx3* expression levels in AVCS), while *Tbx5* was not deregulated. This suggests that the Tbx3/Tbx5 balance model does not hold for the AV node. Several other transcription factors have been implicated in AVCS gene regulation, which together with Tbx3 and Tbx5 form a complex gene regulatory network that has not yet been fully elucidated (*van Eif et al., 2019*). Further studies are required to unravel the mechanisms through which *Tbx3* dose influences the AVCS gene regulatory network and PR interval or QRS duration.

A major limitation of massive parallel reporter assays like STARR-seq is that DNA fragments are tested for regulatory potential episomally and out of genomic context (*Arnold et al., 2013*), lacking the intricate relationship between DNA, histones and long range chromatin interactions (*Gallagher and Chen-Plotkin, 2018*; *Inoue and Ahituv, 2015*). Intrinsic sequence activity or transcription factor binding to motifs within regions endogenously repressed or residing in closed chromatin might activate reporter gene expression when tested in episomal assays. This potentially yields multiple false positive sequences, that is transcriptionally active sequences not relevant for endogenous gene regulation (*Inukai et al., 2017*). Furthermore, we used the reporter vector described in *Arnold et al., 2013* to generate our libraries. A recent study from the same group showed that the bacterial plasmid origin-of-replication within this vector acts as a conflicting core-promoter, thereby causing confounding false-positives and –negatives (*Muerdter et al., 2018*). In addition, the relatively high basal activity of the super core promoter (SCP1) in this vector potentially decreases the fold-change of signal to input (*Muerdter et al., 2018*). As enhancer activity often depends on its ability to interact with the proper promoter (*Arnold et al., 2017*; *Zabidi et al., 2015*), testing our libraries with the endogenous *Tbx3* promoter could yield more relevant RE candidates. The weak overlap of STARR-seq regions with EMERGE and AVCS-ATAC-seq regions illustrates that a subset of the STARR-seq-identified active regions might not be transcriptionally relevant for *Tbx3* expression in vivo. Indeed, the correlation is low between luciferase reporter activity and STARR-seq activity of candidate sequences from the murine genome selected based on either strong epigenomic hallmarks (EMERGE/ChIP-seq/ATAC-seq; high luciferase-/low STARR-seq activity) or strong STARR-seq signal (mSTARR_SG4; high STARR-seq-/low luciferase activity) (*Figure 2—figure supplement 2*). Nevertheless, given the scarcity of alternative epigenomic datasets derived from human AVCS tissue, STARR-seq on the human *TBX3* locus provides a useful albeit somewhat more supportive tool to identify REs.

The lack of readily available AVCS cell lines prompted us to transfect our STARR-seq libraries in COS-7 cells, a monkey kidney-derived fibroblast cell line. Although seemingly irrelevant for assessing regulatory activity of potential AVCS-involved REs, the high transfection efficiency of COS-7 cells allowed us to screen multiple libraries on a large scale and acquire sufficient coverage. The inactivity of a cardiac-specific gene program in COS-7 cells furthermore allowed us to specifically assess the response of genomic fragments to the cotransfected SG4- and Tcf-factors. Our observation that binding motifs for these factors are specifically enriched in AVCS cardiomyocytes, indicated these factors may be involved in regulation of the AVCS gene program. Motif enrichment in AVCS-specific accessible regions was performed by matching AVCS-specific accessible regions to motifs present in the HOMER database. Such databases have their limitations, as they are incomplete and possibly inaccurate as they derive their input from experimental procedures that are condition- and cell type-dependent. Nevertheless, the observed enrichment supported previous studies implicating Bmp-signalling (Smads), Gata4/6 and Wnt-signalling (Tcf) in AVCS patterning (*Stefanovic et al., 2014*; *Gillers et al., 2015*).

Several of the tested risk alleles (e.g. hRE8, hRE9, hRE18) showed a discordant direction of effect between basal activity in COS-7/HL-1 and SG4-/Wnt-stimulated activity. This observation suggests that potentially multiple factors act on these variant REs, with SG4- or Tcf factors counteracting the effect of regulatory networks active in COS-7 or HL-1 cells. Although we could not identify specific TF binding motifs affected by the respective SNPs in these REs, studying TF occupancy in more detail specifically in AVCS cells could elaborate on the precise mechanisms through which these variant REs act. By focusing on SG4- and Wnt-responsive REs, we potentially overlook a proportion of relevant sequences involved in the regulation of *Tbx3* expression, as multiple other factors are involved in AVCS development (*Bhattacharyya and Munshi, 2020*; *Park and Fishman, 2017*; *van Eif et al., 2019*). Recent and ongoing advantages in the generation of (sinoatrial/atrioventricular) nodal-like cells derived from human pluripotent stem cells (*Birket et al., 2015*; *Jung et al., 2014*; *Protze et al., 2017*; *Rimmbach et al., 2015*) will provide a promising tool to study the regulation of genes involved in AVCS development and the function of candidate REs in a considerably more relevant cell type.

# Materials and methods

## Key resources table

| Reagent type (species) or resource | Designation | Source or reference | Identifiers | Additional information |
|---|---|---|---|---|
| Cell line (*Chlorocebus aethiops*) | COS-7 | ATCC | RRID:CVCL_0224 | Fibroblast cell line |
| Cell line (*Mus musculus*) | HL-1 | William C. Claycomb (*Claycomb et al., 1998*) | RRID:CVCL_0303 | Cardiomyocyte cell line |
| Genetic reagent (*Mus musculus*) | $Tbx3^{\Delta VR1/+}$; $Tbx3^{\Delta VR1/\Delta VR1}$ | This paper | | See Materials and methods, section 'TALEN/CRISPR/Cas9-mediated genome editing of VR1 and VR2' |
| Genetic reagent (*Mus musculus*) | $Tbx3^{\Delta VR2/+}$; $Tbx3^{\Delta VR2/\Delta VR2}$ | This paper | | See Materials and methods, section 'TALEN/CRISPR/Cas9-mediated genome editing of VR1 and VR2' |
| Genetic reagent (*Mus musculus*) | $Tbx^{459M16-GFP/+}$ | This paper | | See Materials and methods, section 'BAC modification and immunohistochemistry' |
| Recombinant DNA reagent | pSTARR-seq_human | *Arnold et al., 2013* | Addgene; Plasmid #71509 | |
| Commercial assay or kit | Nextera DNA library prep kit | Illumina | FC-121–1030 | |
| Commercial assay or kit | NEBNext DNA Library Preparation Kit | New England Biolabs | NEB E6000S | |

*Continued on next page*

*Continued*

| Reagent type (species) or resource | Designation | Source or reference | Identifiers | Additional information |
|---|---|---|---|---|
| Commercial assay or kit | NEBNext Multiplex Oligos for Illumina | New England Biolabs | NEB E7335S | |
| Software, algorithm | EMERGE | *van Duijvenboden et al., 2016* | | |

## SNP plotting and human Hi-C analysis

Genome-wide significant SNPs associated with PR interval and QRS duration were extracted from *van der Harst et al., 2016* and *van Setten et al., 2018*. Within genomic loci marked by genome-wide significant SNPs ($p<5*10^{-8}$), SNPs with a p-value<0.05 were included according to the relevance of sub-threshold SNPs described in ref (*Wang et al., 2016*). We also included common variants in high linkage disequilibrium (LD; $r^2 >0.5$) with the top SNPs for both variant regions in our analysis (*Machiela and Chanock, 2015*). SNPs were plotted on the human genome (hg19) using the UCSC Genome Browser (*Kent et al., 2002*). Hi-C data of the human *TBX3* locus and resulting TAD boundaries were obtained from GM12878 human lymphoblastoid cells (*Rao et al., 2014*).

## ATAC-seq on Tbx3+ AVCS cardiomyocytes

ATAC-seq on FACS-purified E12.5-E14.5 $Tbx3^{Venus/+}$ AVJs (*van Eif et al., 2019*) was performed and analyzed as described in *Buenrostro et al., 2013*. In short, $Tbx3^{Venus/+}$ hearts were dissected from embryonic day (E) 12.5-E14.5 embryos and enriched for AV conduction system tissues by microdissection. Single-cell suspensions were obtained using 0.05% Trypsin/EDTA (ThermoFisher Scientific, 25300–054). Cells were sorted on a FacsAria flow cytometer (BD biosciences) and gated to exclude debris and cell clumps, and sorted for $Venus^+$ cells. Approximately 75.000 $Venus^+$ cells were collected and used as input for ATAC-sequencing, which was performed as previously described (*Buenrostro et al., 2013*). The library was sequenced (paired-end 125 bp) and data was collected on a HiSeq2500. ATAC-seq data is deposited in the GEO database under accession number GSE121464.

## STARR-seq library preparation

STARR-seq was performed as described previously (*Arnold et al., 2013*). In short, BACs spanning the entire human and murine *TBX3* locus were used to generate human and murine STARR-seq libraries. BACs were equimolarly pooled, approximately 15 µg of pooled BAC DNA was sheared by sonication (Amp 20%, 2 × 15 s) and fragments of approximately 500–1000 bp were selected by gel extraction followed by cleanup using the Qiaquick Gel extraction kit (Qiagen; 28704). 1.5 µg of size-selected DNA fragments was used as input for library preparation using the NEBNext DNA Library Preparation Kit (NEB; E6000S) and NEBNext Multiplex Oligos for Illumina (NEB E7335S) according to manufacturer's protocol. Recombination of adapter-ligated DNA fragments, transformation and library DNA isolation was performed as described (*Arnold et al., 2013*).

## STARR-seq transfection

$15*10^6$ COS-7 cells were cultured in DMEM (ThermoFisher Scientific; 31966–021) supplemented with 10% FBS (ThermoFisher Scientific; 10270–106) and 1% Pen/Strep (ThermoFischer Scientific; 15070–063). Cells were transfected with 150 µg library DNA and 75 µg of pcDNA (control), SG4 (15 µg Smad1, 15 µg Smad4, 30 µg Alk3 and 15 µg Gata4 expression vectors) or Wnt (15 µg Tcf4 expression vector, 60 µg pcDNA, and 0.4M LiCl) using Polyethylenimine 25 kDa (PEI; Sigma-Aldrich; 408727) in a ratio of 1:3 (DNA:PEI). Medium was refreshed 6 hr after transfection. 48 hr after transfection, total RNA was isolated using the Qiagen RNeasy maxi prep kit (Qiagen; 75162). polyA+ RNA was isolated using Dynabeads Oligo-dT$_{25}$ (ThermoFisher Scientific; 61002) and treated with Ambion turboDNase (ThermoFisher Scientifc; AM2238) for 30 min at a maximum concentration of 150 ng/µl. RNA was purified using the Qiagen RNeasy MinElute kit (Qiagen; 74204). First-strand cDNA synthesis and library preparation was performed as described (*Arnold et al., 2013*). Library quality and concentration were assessed by Bioanalyzer (Agilent) and KAPA Library Quantification

Kit (KAPA Biosystems; KK4824). Libraries were pooled equimolarly and sequenced on an Illumina MiSeq (PE150).

## STARR-seq data analysis

Paired-end sequence reads were mapped to mm9 (mSTARR) or hg19 (hSTARR) using BWA (*Li and Durbin, 2009*). For visualization purposes, bam files were converted to bigwig files using bamCoverage (*Ramírez et al., 2016*) and visualized using the UCSC genome browser (*Kent et al., 2002*). We compared bam files of m/hSTARR_SG4/Wnt to m/hSTARR_control using bamCompare (*Ramírez et al., 2016*) with bin sizes of 50 bases to compute the log2 of the number of reads ratio. Bins were filtered to exclude regions with a read count <75. Flanking bins were merged using mergeBED (overlaps on either strand; maximum distance between features: 0) (*Quinlan and Hall, 2010*). Merged bins with a log2 fold change of >0.585 (fold change >1.5) were included for further analysis. For overlap analysis, bed files were intersected using BEDTools (*Quinlan and Hall, 2010*) with a minimum interval overlap of 50 bp.

## EMERGE enhancer selection

Putative mouse cardiac enhancers were predicted by EMERGE (*van Duijvenboden et al., 2016*) using a total of 116 selected functional genomic datasets. These included ChIP-seq data of enhancer-associated histone modification marks and transcription factor binding sites, and chromatin accessibility data as assessed by DNAseI-hypersensitivity and ATAC-seq, derived predominantly from cardiac cells. By assigning weight to all selected datasets through a logistic regression modeling approach (described in *van Duijvenboden et al., 2016*) using validated heart enhancers as training set, a genome-wide cardiac enhancer prediction track was generated. The same exercise was performed for the human genome, using a total of 70 selected functional genomic datasets. An overview of the datasets used of mouse and human EMERGE enhancer prediction is provided in *Supplementary file 1*-Supplementary Tables 10, 11.

## ATAC-seq and H/mSTARR regions motif analysis

The ATAC-seq dataset on FACS-purified E12.5-E14.5 *Tbx3*$^{Venus/+}$ AVJs was utilized to identify accessible regions within the *Tbx3* locus. As genome-wide peak-calling algorithms did not fully call all peaks within the *Tbx3/Tbx5* locus, even with less stringent parameters, we used a local, locus-specific threshold value for the selection of peaks. The entire sequence surpassing the threshold was included and considered as potential RE. A similar approach was used to select ventricular ATAC-seq regions (*van Duijvenboden et al., 2019*), and a random selection of 67 regions from this set and the 67 accessible AVJ regions were used as input for the HOMER motif analysis tool (*Heinz et al., 2010*) to identify AVCS-specific enriched sequence motifs within the *Tbx3* locus.

Murine/human STARR-seq regions with fold change of stimulation over control of >1.5 (see: *STARR-seq data analysis*) were used as input for the HOMER motif analysis tool (*Heinz et al., 2010*). Potential enrichment against genome background was checked for all known motifs in the JASPAR database (*Fornes et al., 2020*). To allow for comparison of motif enrichment between different datasets, we used 150 randomly selected active regions for hSTARR_SG4 and hSTARR_Wnt, and 181 randomly selected active regions for mSTARR_SG4 and mSTARR_Wnt as input. Input regions were defined as the complete genomic region surpassing the >1.5 fold change threshold for activation over control.

## Cloning of RE candidates

Murine and human regions selected based on either STARR-seq activity or EMERGE/ATAC-seq prediction for validation were amplified from respective BAC DNA (used for STARR-seq library preparation) in which they reside. Major and minor haplotypes of identified RE candidates were amplified from human DNA from individuals within the CONCOR-genes database. PCR amplification of RE sequences was performed using Q5 Hot Start High-Fidelity DNA Polymerase (NEB; M0493S). Primer sequences are listed in *Supplementary file 1*-Supplementary Tables 7-9. Amplified sequences were ligated in the *XcmI* site of a modified pGL2-SV40 vector (Promega) using T4 DNA Ligase (ThermoFisher Scientific; 15224–090). After transformation, plasmid DNA was isolated using the PureLink

HiPure Plasmid Midiprep kit (ThermoFisher Scientific; K210005) and purified by phenol/chloroform extraction.

## Cell culture

HL-1 cells (RRID:CVCL_0303; 2.5*10$^6$ cells/plate) were grown in 12-well plates in Claycomb medium (Sigma-Aldrich, 51800C) supplemented with chemically defined HL-1 FBS substitute (Lonza, 77227), Glutamax (ThermoFisher Scientific, 35050–061) and Pen/Strep (ThermoFisher Scientific, 15070–063). COS-7 cells (RRID:CVCL_0224; 1.5*10$^6$ cells/plate) were grown in 12-well plates in DMEM (Thermo-Fisher Scientific, 31966–021) supplemented with 10% FBS (ThermoFisher Scientific, 10270–106) and Pen/Strep (ThermoFisher Scientific, 15070–063). Both HL-1 and COS-7 cell lines were routinely tested negative for mycoplasma contamination. HL-1 cells and COS-7 cell lines were easily distinguished based on cellular morphology and contractility (HL-1). Neither HL-1 or COS-7 is found in the database of commonly misidentified cell lines that is maintained by the International Cell Line Authentication Committee.

## Transient transfection and luciferase assays

HL-1 cells were transfected with plasmid DNA using Lipofectamine 3000 (ThermoFisher Scientific, L3000-015). 1 µg plasmid DNA was transfected using 4 µL P3000 reagent and 3 µL Lipofectamine. COS-7 cells were transfected using Polyethylenimine 25 kDa (PEI; Sigma-Aldrich; 408727) at a 1:3 ratio (DNA:PEI). 1 µg of reporter plasmid DNA was transfected with either 500 ng of pcDNA3.1(+) (ThermoFisher Scientific, V790-20) as control, or 100 ng Smad1, 100 ng Smad4, 200 ng Alk3 and 100 ng Gata4 expression vectors (SG4) and 100 ng Tcf4, 400 ng pcDNA, and 0.4M LiCl (Wnt). Medium was refreshed 6 hr after transfection. Cells were lysed 48 hr after transfection using Renilla luciferase assay lysis buffer (Promega, E291A-C). Luciferase activity measurements were performed in duplo using a GloMax Explorer (Promega, GM3500) with 100 ul D-Luciferin (p.j.k., 102111). Measurements were performed by a 1 s delay and 5 s of measurement per sample.

## Analysis of disrupted/de novo created TF binding motifs in variant REs

Of the variant RE candidates of which the risk allele affected luciferase activity, we took the variant nucleotide and 10 nucleotides both up- and downstream. These 21 bp sequences were used as input for transcription factor binding motif recognition in JASPAR (*Fornes et al., 2020*) using default settings.

## BAC modification and immunohistochemistry

The modification and analysis of BACs RP24-89K7-GFP, RP23-143N21-GFP and RP23-366H17-GFP has been previously described (*Horsthuis et al., 2009*),(*van Weerd et al., 2014*). BAC RP24-459M16 was obtained from a C57BL/6J mouse BAC library (CHORI, BACPAC Resources) and modified to generate 459M16-GFP. To this end, a GFP reporter gene cassette coupled to the minimal heat shock promoter 68 (hsp68-GFP) was inserted at 81 kb from the 5'-end of BAC 459M16 following the two-step BAC modification protocol as described in *Gong et al., 2002*. Modified 459M16-GFP DNA was purified using the Nucleobond PC20 kit (Machery Nagel) and injected in pronuclei of FVB/N mice. Microinjected zygotes were implanted in pseudo-pregnant females (timepoint E0.5) and embryos were isolated at E13.5. GFP$^+$ embryos were fixed in 4% paraformaldehyde in PBS for 4 hr and processed for immunohistochemistry, as described in *van Weerd et al., 2014*. Three independent GFP$^+$ founders were analysed for GFP expression.

## TALEN/CRISPR/Cas9-mediated genome editing of VR1 and VR2

For the deletion of VR1 from the mouse genome (Tbx3$^{\Delta VR1}$), two genomic sites flanking were targeted by TALEN. TALEN target sequences were designed with TALENT2.0 (https://tale-nt.cac.cornell.edu/) and assembled according to the Golden Gate cloning protocol (*Cermak et al., 2011*). Genomic coordinates and RVD sequences of TALEN target sites are listed in *Supplementary file 1*-Supplementary Table 4 (*Carlson et al., 2012*).

Assembled RVDs were cloned in the pC-GoldyTALEN and RCIscript-GoldyTALEN (*Carlson et al., 2012*) destination vector to generate mRNA expression plasmids. TALEN mRNA was in vitro transcribed by linearization of the expression plasmid with SacI at 37˚C for 3 hr, followed by transcription

of the linearized DNA using the T7 mMessage Machine kit (ThermoFisher Scientific; AM1344). 20 ng/μl of TALEN mRNA was injected in the cytoplasm of one-cell embryos of FVB/N mice. Positive founders were identified by PCR with primers flanking the 69 kb deletion and subsequent Sanger sequencing. The genomic deletion in positive founders was backcrossed for at least two generations with wildtype FVB/N mice before further experiments.

To delete VR2 from the mouse genome ($Tbx3^{\Delta VR2}$), oligonucleotides for the generation of two single guide RNAs (sgRNAs) targeting two genomic sites flanking VR2 in the mouse genome were designed using the online tool ZiFit Targeter (*Sander et al., 2010*). Target site sequences are listed in *Supplementary file 1*-Supplementary Table 5. sgRNA oligonucleotides were ligated into *Bsal*-digested pDR274 using T4 DNA Ligase (Invitrogen). sgRNA (pDR274) and Cas9 (MLM3613) expression vectors were linearized with *DraI* (NEB, R0129S) and *PmeI* (NEB, R0560S), respectively. sgRNA and Cas9 in vitro transcription was performed using the MEGAshort T7 kit (Life Technologies, AM1354) and mMessage mMachine T7 Ultra kit (Life Technologies, AM1345), respectively, followed by purification using the MEGAclear kit (Life Technologies, AM1908). 10 ng/μl sgRNA and 25 ng/μl Cas9 mRNA was micro-injected into the cytoplasm of one-cell embryos of FVB/N mice. The genomic deletion in positive founders was backcrossed for at least two generations with wildtype FVB/N mice before further experiments.

## In situ hybridization

E13.5 embryos were isolated in ice cold PBS and fixated overnight in 4% paraformaldehyde in PBS, dehydrated in a graded ethanol series, embedded in paraplast and sectioned at 10 μm. Non-radioactive in situ hybridization on sections was performed as previously described (*Moorman et al., 2001*) using mRNA probes for the detection of *Tbx3*, *Tbx5* and *Hcn4*. Stained sections were examined with a Zeiss Axiophot microscope and photographed with a Leica DFC320 Digital Camera.

## Quantitative expression analysis

For the quantification of expression levels, E13.5 embryos from $Tbx3^{+/+}$, $Tbx3^{VR1/+}$, $Tbx3^{VR2/+}$, $Tbx3^{VR1/VR1}$ and $Tbx3^{VR2/VR2}$ were isolated in ice cold PBS and liver, limbs, lungs, AVJs and SANs were isolated by microdissection. Total RNA was isolated by the ReliaPrep RNA Tissue Miniprep System (Promega; Z6112) according to manufacturer's protocol. First-strand cDNA was synthesized with 500 ng total RNA as input using SuperScript II Reverse Transcriptase (ThermoFischer Scientific; 18064022). Quantitative real-timePCR was performed in technical duplicates using the Roche Light-Cycler 480 system. The oligonucleotide sequences for the amplification of the different amplicons are listed in *Supplementary file 1*-Supplementary Table 6. The relative start concentration was calculated as described in *Ruijter et al., 2009*. Expression values of the different amplicons were normalized to the geomean of *Hprt* and *Eef2* expression (liver, limbs, lungs), *Tbx2* expression (AVJ) and *Isl1* expression (SAN). Expression values in tissues of $Tbx3^{VR1/+}$, $Tbx3^{VR2/+}$, $Tbx3^{VR1/VR1}$ and $Tbx3^{VR2/VR2}$ mice were compared to those of wild-type littermates ($Tbx3^{+/+}$).

## Recording of in vivo and ex vivo electrocardiograms

$Tbx3^{+/+}$, $Tbx3^{\Delta VR1/\Delta VR1}$ and $Tbx3^{\Delta VR2/\Delta VR2}$ mice (male, 2–4 months old) were anesthetized with 4% Isoflurane (Pharmachemie B.V.) and maintained at 1.5–2.0%. Electrodes were placed at the right (R) and left (L) armpit and the left groin (F) and an electrocardiogram (ECG) was recorded (PowerLab 26T; AD-Instruments, Colorado Springs, CO, USA) for a period of 5 min. ECG parameters were determined in Lead II (L-R) based on the last 30 s of the recording.

To record ECGs ex vivo the adult mice were sacrificed by $CO_2$ (1 L/min inflow) and cervical dislocation. The heart was rapidly excised, cannulated, mounted on a Langendorff perfusion set-up as described previously (*Boukens et al., 2013*). Neonatal mice (ND0-1) were sacrificed by 4% Isoflurane and cervical dislocation. Hearts were isolated and superfused with HEPES-buffered Tyrode's solution (containing in mmol/L: 140 NaCl, 5.4 KCl, 1.8 CaCl2, 1.0 MgCl2, 5.5 glucose and 5.0 HEPES) at a temperature of 36 ± 0.2°C; pH was set to 7.4 with NaOH. During perfusion the heart was submerged and electrodes were placed at the right (R) and left (L) side of the base of the heart and at the left side of the apex (F) at a 5 mm distance. Lead II was used to determine ECG parameters. ECG parameters both in vivo and ex vivo for $Tbx3^{\Delta VR1/\Delta VR1}$ and $Tbx3^{\Delta VR2/\Delta VR2}$ mice were compared to those of wild-type littermates ($Tbx3^{+/+}$).

## Acknowledgements

We would like to thank Corrie de Gier-de Vries for help with the in-situ hybridizations and Connie Bezzina and Doris Skoric-Milosavljevic for supplying human DNA samples for the amplification of RE haplotypes. This work was supported by the Dutch Heart Foundation/CVON project CON-COR-genes, Fondation Leducq (14CVD01) and Dutch Heart Foundation grant COBRA3 (2012T091) to VMC, and Dutch Heart Foundation grant 2016T047 to BJB.

## Additional information

### Funding

| Funder | Grant reference number | Author |
| --- | --- | --- |
| Dutch Heart Foundation | CVON project CONCOR genes | Vincent M Christoffels |
| Dutch Heart Foundation | COBRA3 | Vincent M Christoffels |
| Fondation Leducq | 14CVD01 | Vincent M. Christoffels |
| Dutch Heart Foundation | 2016T047 | Bastiaan J Boukens |

The funders had no role in study design, data collection and interpretation, or the decision to submit the work for publication.

### Author contributions

Jan Hendrik van Weerd, Conceptualization, Data curation, Formal analysis, Validation, Investigation, Visualization, Writing - original draft, Writing - review and editing; Rajiv A Mohan, Conceptualization, Formal analysis, Validation, Investigation; Karel van Duijvenboden, Conceptualization, Software, Formal analysis, Investigation, Methodology; Ingeborg B Hooijkaas, Vincent Wakker, Formal analysis, Validation, Investigation; Bastiaan J Boukens, Conceptualization, Validation, Investigation, Methodology; Phil Barnett, Conceptualization, Formal analysis, Supervision, Writing - original draft, Writing - review and editing; Vincent M Christoffels, Conceptualization, Data curation, Formal analysis, Supervision, Funding acquisition, Validation, Investigation, Visualization, Methodology, Writing - original draft, Project administration, Writing - review and editing

### Author ORCIDs

Jan Hendrik van Weerd (iD) https://orcid.org/0000-0001-7955-8477
Rajiv A Mohan (iD) https://orcid.org/0000-0002-3622-1759
Vincent M Christoffels (iD) https://orcid.org/0000-0003-4131-2636

### Ethics

Animal experimentation: Animal care and experiments were conducted in accordance with guidelines from the European Union, Dutch government, and Amsterdam University Medical Centers, and approved by the Animal Experimental Committee of the Amsterdam University Medical Centers and the Central Committee Animal Experiments (CCD) under AVD1180020172348.

### Decision letter and Author response

Decision letter https://doi.org/10.7554/eLife.56697.sa1
Author response https://doi.org/10.7554/eLife.56697.sa2

## Additional files

### Supplementary files

• Supplementary file 1. Supplementary Tables 1—11. Supplementary Table 1. Transcription factor binding motif enrichment in genomic regions accessible in E14.5 atrioventricular junctions. Transcription factor binding motif enrichment in 67 genomic regions accessible in E14.5 atrioventricular

junctions (AVJ) or E14.5 ventricles. Enrichment was compared to enrichment in a number of random control sequences. Enrichment is depicted as total number of regions with the motif and the percentage of the total number of tested regions. P-values depict statistical significance of enrichment. Supplementary Table 2. Transcription factor binding motif enrichment in active STARR-regions. Transcription factor binding motif enrichment in genomic regions within both mouse (A) and human (B) *TBX3* locus that respond to Smad/Gata (SG4) or Tcf (Wnt) factors as demonstrated by STARR-seq. Enrichment of binding motifs for Smad/Gata- and Tcf-factors is depicted. Enrichment was compared to enrichment in a number of random control sequences. Enrichment is depicted as total number of regions with the motif and the percentage of the total number of tested regions. P-values depict statistical significance of enrichment. Supplementary Table 3. Murine RE candidates within VR1 and VR2. Murine RE candidates selected based on accessible chromatin in AV junction cardiomyocytes (ATAC_AVJ), EMERGE prediction, or mSTARR_SG4/Wnt activity. Overlap is depicted of candidate regions with ChIP-seq peaks from publicly available datasets for Gata4 (differentiated cardiomyocytes) (*Luna-Zurita et al., 2016*), Hand2 (E10.5 heart) (*Laurent et al., 2017*) and H3K27ac (E11.5 heart) (*Nord et al., 2013*). Supplementary Table 4. Human RE candidates within VR1 and VR2. Human RE candidates selected based on mouse RE homology (mm9 liftover), EMERGE prediction, or hSTARR_SG4/Wnt activity. Regions overlapping GWAS SNPs (SNP id, trait association and respective p-value are listed) were selected for further analysis. Supplementary Table 5. TALEN/CRISPR target sequences for deletion of VR1 and VR2 from the murine genome Supplementary Table 6. qPCR primer sequences Supplementary Table 7. Overview of BACs used for the generation of STARR-seq libraries Supplementary Table 8. Primer sequences for the amplification of active STARR-seq regions for validation by luciferase reporter assay Supplementary Table 9. Primer sequences for the amplification of RE candidate fragments Supplementary Table 10. Mouse functional genomic datasets included in merge as possible predictors Supplementary Table 11. Human functional genomic datasets included in merge as possible predictors.

• Transparent reporting form

### Data availability

Sequencing data have been deposited in GEO under accession codes GSE121464 and GSE145257.

The following dataset was generated:

| Author(s) | Year | Dataset title | Dataset URL | Database and Identifier |
|---|---|---|---|---|
| van Weerd JH, van Duijvenboden K, Christoffels VM | 2020 | STARR-seq of human and mouse Tbx3 locus | http://www.ncbi.nlm.nih.gov/geo/query/acc.cgi?acc=GSE125257 | NCBI Gene Expression Omnibus, GSE125257 |

The following previously published dataset was used:

| Author(s) | Year | Dataset title | Dataset URL | Database and Identifier |
|---|---|---|---|---|
| Mohan RA, van Weerd JH, van Duijvenboden K, Christoffels VM | 2019 | Tbx3 governs a transcriptional program to maintain atrioventricular conduction system form and function [ATAC-seq] | https://www.ncbi.nlm.nih.gov/geo/query/acc.cgi?acc=GSE121464 | NCBI Gene Expression Omnibus, GSE121464 |

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
