## [Decision Letter]

**Acceptance summary:**

The manuscript presents an integrated approach for investigating genetic variation identified by GWAS, traveling from human genetics through informatics and genomic analysis to validation in a mouse model. This approach may be broadly applicable. Equally interesting are the specific findings concerning gene regulation at the TBX3 locus, including the identification of several enhancers, at least one of which is shown to be required in vivo.

**Decision letter after peer review:**

Thank you for submitting your article "Trait-associated noncoding variant regions affect *TBX3* regulation and cardiac conduction" for consideration by *eLife*. Your article has been reviewed by three peer reviewers, including Ivan P Moskowitz as the Reviewing Editor and Reviewer #1, and the evaluation has been overseen by Didier Stainier as the Senior Editor The following individual involved in review of your submission has agreed to reveal their identity: Tony Firulli (Reviewer #2).

The reviewers have discussed the reviews with one another and the Reviewing Editor has drafted this decision to help you prepare a revised submission.

Summary:

Hendrik van Weerd et. al is a very good manuscript detailing a functional genomics approach to understanding regulation of the TBX3 gene and investigating GWAS variants for cardiac rhythm traits at the TBX3 locus. The paper includes an interrogation of common non-coding genomic variants upstream and in cis with the T-box transcription factor TBX3 associated with changes in cardiac conduction velocity.

The paper excels in several ways. First, it provides a comprehensive approach to travel from trait-associated variants to functional investigation of the gene regulatory consequences of specific variants, attempting to link them to phenotypic consequences. Second, it advances our understanding of the cis-regulatory landscape at TBX3 and reveals a likely molecular mechanism for GWAS signals at the TBX3 locus – specifically, variation of cis-regulatory element function affecting TBX3 expression in conduction tissues. Third, it considers and tackles some basic difficulties in understanding cardiac conduction system GWAS – the lack of human expression data for cardiac conduction tissues and therefore the lack of eQTL data for such tissues implicated in cardiac rhythm control. The mouse – human pipeline presented provides a potential roadmap for future studies in the CCS or in other tissues that lack human expression data.

Revisions:

The specific comments are meant to clarify the manuscript. The generation of new data is not necessary for the revision. The following issues should to be clarified in a revision manuscript, either by analysis of current data or by clarification of the text and/or figures:

1) In-vivo functional analysis is not of SNPs themselves but of a much larger enhancer deletion. This caveat should be discussed with regards to the molecular mechanism underlying the GWAS association caused by SNPs.

2) Some of the observed molecular and functional effects of SNPs or enhancer knockouts are unexpected and are left without a mechanistic explanation. For example, analysis of the in-vivo effects of removal of VR2 generated results for Tbx3 expression and for functional conduction measures that are opposite of expectations. Tbx3, a repressor of fast conduction channels, was upregulated in conduction tissues, generating the expectation that conduction would be slowed in affected regions. However, a decreased PR interval, indicative of more rapid conduction speed, was observed. Can the authors explain why this is the case? At the very least the authors should discuss the unexpected nature of this observation and offer a plausible explanation.

3) The Motif data presented in a few of the figures may not be accurate. It is clear that the VR2 is functional and published DNA occupancy data shows associated DNA binding from defined TFs within the RE's; however, the sequence consensus shown for the bHLH factors (sans MyoD) do not contain E-boxes, *SOX9* and SMAD Motifs reported also appear non-canonical. Discussing this at some level is warranted.

4) ATAC-seq was performed in embryonic hearts and traits were from adult humans. This comparison may underestimate the overlap were ATAC to be done on adult AVC cells. This caveat should be discussed.

5) The authors interrogated the human TBX3 locus for TAD structure and the location of SNPs associated with conduction traits; they identified candidate REs in the AVC by performing AVC ATAC-seq, which should provide an excellent resource for the field.

How were the 67 candidate REs at the Tbx3 locus defined relative to the entire dataset?

How specific are the identified elements for the AVC as opposed to non-AVC myocardium?

The authors show but do not discuss the comparison between the AVC versus ventricular ATAC across the Tbx3 locus. How many are AVC-specific? It looks like many are shared.

Did the author's attempt a differential ATAC analysis across the locus?

6) The STARR analysis was performed in COS cells to identify regions that activate transcription based on specific TFs previously defined as important for AVC gene expression. As the authors indicate, there is no AVC cell line, requiring use of an unrelated line. None-the-less, the use of COS cells is a significant caveat, and should be discussed clearly in the Discussion. The TF enrichment in this context is a therefore a self-fulfilling prophecy.

The number of human and mouse STARR identified candidates, as well as their overlap, should be described up front.

7) The description of mouse/human homology for VR1 and VR2 should be described in more detail.

8) RE candidate selection is described in Fg3a using a hierarchical approach in schematic form.

It should be possible to use the actual locus, display the individual datasets, and overlay them to present a more accurate description of the chosen REs at the locus.

Figure 3B appears to display data already published. Were any other candidates tested in-vivo?

9) Several of the candidate risk alleles, examined in luciferase activity in Figure 4C and d, show discordant effects on basal activity and TF-stimulated activity. This should be described and considered. Do the variants alter TF binding sites of known activator or repressor TFs that would help explain this discordance? Do any of the analyzed variants alter TF binding sites of the TF analyzed, which may explain their altered regulatory response to TF expression?

10) In Figure 6, the schematic shows the location of the TALEN cuts. The authors should delineate VR1 and VR2, as in Figure 5.

11) Figure 1C do the authors have insight as to why the Motif consensus for HAND2, TCF3, and TCF4 do not contain a clear E-box motif such as shown for MyoD? The *SOX9* Motif also seems out of place with the canonical consensus: CCTTGAG. SMAD 3&4 likewise are reported to bind CAGAC, CAGCC as well as the 5-bp consensus sequence GGC(GC)|(CG). What is shown in the motif is very different and some discussion of this is warranted there is not confidence that the aforementioned transcription factors would bind these sequences robustly. There is simply low confidence that the algorithm is pulling accurate motif data and the authors should try to address this in the text.

12) Figure 2 same issue with consensus data in 2G (note the motif name and the motif do not line up correctly).

13) Figure 3 some clarity in the narration would be helpful. Paragraph one of subsection “Identification of variant REs within VR1 and VR2” discuss 9 candidate REs in VR1 and 13 in VR2. Are these regions supposed to be identifiable in Figure 3A (as is the assumption)? The figure shows 6 candidate REs which resolve to 3 with SNP overlaps comparing the figure to the narration is confusing.

14) Figure 6C shows only Tbx3 expression Tbx5 and Med13l expression narrated as being in Figure 6C. Panel G would benefit from slightly enlarged images.

15) The criteria and datasets used to define the EMERGE regions is not found in this paper nor is there any citation listing where these regions came from. Please clarify.

16) What was the criteria used to identify the 9 and 13 (as listed in the text) candidate regulatory elements in VR1/2? Supplementary Table 4 lists only 20 candidate regulatory elements (7 and 13). What happened to hRE1 and hRE2 to exclude them from downstream analysis and inclusion in the table? Were other sites excluded as well? It is also unclear from Figures 2H-K and Figure 3A, and Supplementary Tables 3 and 4 what was used to define the candidate regulatory elements examined in greater detail in Figures 4, Figure 3—figure supplement 1 and Figure 4—figure supplement 1. For example, hRE13 was not examined in Figures 4C/D or Figure 4—figure supplement 1. Overall, I really like the multi-factor approach to defining candidates; however, the key filtering steps taken to get the final list of candidates needs to be more clearly defined.

---

## [Author Response]

Revisions:The specific comments are meant to clarify the manuscript. The generation of new data is not necessary for the revision. The following issues should to be clarified in a revision manuscript, either by analysis of current data or by clarification of the text and/or figures:1) In-vivo functional analysis is not of SNPs themselves but of a much larger enhancer deletion. This caveat should be discussed with regards to the molecular mechanism underlying the GWAS association caused by SNPs.

We agree and have elaborated in the Discussion: “Our current findings further substantiate regulatory redundancy in vivo. […] Therefore, we set out to assess the effect of deletion of the entire variant regions within the TBX3 locus rather than assessing the effect of individual SNPs, and found that the two TBX3 risk loci harbor multiple active REs whose activity is affected by their respective risk alleles.”

2) Some of the observed molecular and functional effects of SNPs or enhancer knockouts are unexpected and are left without a mechanistic explanation. For example, analysis of the in-vivo effects of removal of VR2 generated results for Tbx3 expression and for functional conduction measures that are opposite of expectations. Tbx3, a repressor of fast conduction channels, was upregulated in conduction tissues, generating the expectation that conduction would be slowed in affected regions. However, a decreased PR interval, indicative of more rapid conduction speed, was observed. Can the authors explain why this is the case? At the very least the authors should discuss the unexpected nature of this observation and offer a plausible explanation.

We agree the result was unexpected. Reduction of Tbx3 also results in PR shortening (Frank et al., 2012), suggesting that both more and less than normal Tbx3 causes apparent faster conduction through the AV conduction system. While several channels and gap junction subunits that are also targets of Tbx3 have been indicated to dose dependently contribute to conduction (Scn5a/Nav1.5, Gja5/Cx40 etc.), many more genes will be deregulated in the mutant AV conduction system, leading to changes in a complex genetic network with currently unpredictable output. We have discussed this in the Discussion section, also referring to recent work from the Moskowitz lab regarding the interplay between Tbx5 and Tbx3 in the AV bundle.

“Not only Tbx3 insufficiency (Bakker et al., 2008; Frank et al., 2011; Horsthuis et al., 2009) but also over-expression (this study, Burnicka-Turek et al., 2020) can be detrimental, suggesting proper AVCS function depends on strictly balanced Tbx3 levels. […] Further studies are required to unravel the mechanisms through which *Tbx3* dose influences the AVCS gene regulatory network and PR interval or QRS duration.”

3) The Motif data presented in a few of the figures may not be accurate. It is clear that the VR2 is functional and published DNA occupancy data shows associated DNA binding from defined TFs within the RE's; however, the sequence consensus shown for the bHLH factors (sans MyoD) do not contain E-boxes, SOX9 and SMAD Motifs reported also appear non-canonical. Discussing this at some level is warranted.

We agree with the reviewer, and have specified interpretations of the Motif analysis. The confidence of such analyses depends on the quality of motifs within the database, which are experimentally derived from different experiments, in various cell types or developmental stages. As such, scanning sequences for motif enrichment using HOMER or other large scale analysis tools should be interpreted with this in mind. Nevertheless, we used the analysis to obtain a global impression of the factors that could be involved in AVCS Tbx3 regulation, while being aware that the output list is far from comprehensive. We have toned down statements regarding the conclusion of this analysis in the Results and Discussion section, and have specified in the Discussion that we used this analysis mainly to obtain an impression of involved factors, and have discussed the caveats of this analysis. Furthermore, we have removed the motif logos from Figure 1C.

4) ATAC-seq was performed in embryonic hearts and traits were from adult humans. This comparison may underestimate the overlap were ATAC to be done on adult AVC cells. This caveat should be discussed.

We agree with the reviewer that we might underestimate the overlap of GWAS-SNPs with potential REs, as we do not include adult AVC ATAC-seq in our analysis. However, we assume that a large proportion of the potential regulatory elements accessible in adult tissues are also accessible at developmental stages. The embryonic ATAC-seq data is furthermore only one of the filtering steps for selecting candidate REs, as we also use mouse and human STARR-seq, and mouse and human EMERGE prediction, which is based on multiple datasets including some derived from adult mouse tissues and human tissues. Nevertheless, including ATAC-seq data derived from adult Tbx3+ hearts could yield additional regulatory elements that are potentially missed in this configuration and we have added this limitation to the Conclusion.

5) The authors interrogated the human TBX3 locus for TAD structure and the location of SNPs associated with conduction traits; they identified candidate REs in the AVC by performing AVC ATAC-seq, which should provide an excellent resource for the field.How were the 67 candidate REs at the Tbx3 locus defined relative to the entire dataset?How specific are the identified elements for the AVC as opposed to non-AVC myocardium?The authors show but do not discuss the comparison between the AVC versus ventricular ATAC across the Tbx3 locus. How many are AVC-specific? It looks like many are shared.Did the author's attempt a differential ATAC analysis across the locus?

To increase specificity and minimize the risk of unjustly excluding potential accessible regions for downstream analysis, we used a local threshold value for our selection of ATAC-AVJ regions. We observed that, even with less stringent parameters, genome-wide peak-calling algorithms did not fully call all peaks (we are aware that this observation was done by visually inspecting the ATAC-seq track in the murine *Tbx3* locus and that it is therefore subjective). As such, we used a local, locus-specific threshold value for the selection of peaks – the entire sequence surpassing the threshold was included and considered as potential RE. A similar approach was used to select ventricle ATAC-peaks, and a random selection of 67 regions from this set was used for the motif enrichment analysis.

From these 67 accessible fetal AVCS regions, 28 overlapped with accessible adult ventricular regions, indeed a fair share of overlap. We have added this to Figure 1. This large portion of overlap can be explained by the fact that although AVCS and ventricle are distinct cell types, both are types of cardiomyocytes and therefore potentially share multiple transcription factors or regulatory machinery components that bind to their genomic location. We tried performing the analysis on regions only accessible in either AVCS or ventricle, however, the number of input regions was too low and underpowered. Furthermore, we note that the previously identified AVCS-specific enhancers Tbx3-eA and Tbx3-eB are accessible in multiple cell types, including our AVCS and ventricle ATAC-seq datasets, yet show highly specific activity in only the AVCS. Excluding regions that are also accessible in the ventricle dataset, or only focusing on the AVCS-specific regions, would omit such potentially strong RE candidates from further analysis. We have included and expanded the Materials and methods section to elaborate more on the inclusion criteria for ATAC peaks and their downstream analysis.

6) The STARR analysis was performed in COS cells to identify regions that activate transcription based on specific TFs previously defined as important for AVC gene expression. As the authors indicate, there is no AVC cell line, requiring use of an unrelated line. None-the-less, the use of COS cells is a significant caveat, and should be discussed clearly in the Discussion. The TF enrichment in this context is a therefore a self-fulfilling prophecy.The number of human and mouse STARR identified candidates, as well as their overlap, should be described up front.

We have edited the Results section according to the reviewer’s suggestion and moved the description on the number of human and mouse STARR REs, and their overlap, up front. We have revised Figure 2 accordingly.

We agree with the reviewer’s comments regarding the caveat of using the COS7 cell line for the identification of AVCS-specific regulatory elements. However, in the experiments described in this manuscript, the high transfection efficiency of the COS7 cells allowed us to reach high resolution in the STARR-seq output, which requires large numbers of transfected cells. Indeed, due to low transfection efficiencies, STARR-seq in “atrial” HL1 cells failed. The lack of a non-AVCS-myocardium gene program allowed us to specifically focus on the response of regions in the murine/human TBX3 locus to SG4 or Wnt factors.

The TF enrichment for the respective stimuli could indeed be seen as a self-fulfilling prophecy. However, the aim of this exercise was to validate the cutoff for regulatory activity rather than identify motifs specifically present in these regions, in other words: to see if we, with a rather low threshold for regulatory activity of FC>1.5 would still extract relevant sequences while minimizing the change to falsely exclude regions. As such, we wanted to assess whether the active regions we identified with the FC>1.5 threshold would yield relevant sequences, rather than noise caused by inter-experiment variation. Indeed, we found that the active regions in the respective STARR datasets were enriched for binding motifs of the respective factors that we co-transfected the libraries with, indicating the used threshold was justified.

We have revised the Discussion and included some discussion regarding the use of COS-7 cells and the potential caveats. Furthermore, we have revised the Discussion on the interpretation and use of the binding motif enrichment analysis according to the reviewer’s comments, also in line with the question of reviewer 2.

7) The description of mouse/human homology for VR1 and VR2 should be described in more detail.

We have added a track displaying sequence conservation in the mouse and human genomes, respectively, to Figure 5, and have added information regarding the sequence homology between mouse and human VR1 and VR2 in the Results section. VR2 also harbors the AVC-specific RE that was functionally and structurally conserved between mouse and human.

8) RE candidate selection is described in Fg3a using a hierarchical approach in schematic form.It should be possible to use the actual locus, display the individual datasets, and overlay them to present a more accurate description of the chosen REs at the locus.Figure 3B appears to display data already published. Were any other candidates tested in-vivo?

We agree with the reviewer that the use of the actual locus with the respective datasets would provide a more accurate description of the workflow, and, more importantly, of the filtering/inclusion steps at each step. We have discarded the schematic illustration of the former Figure 3 and revised the figure to show the actual datasets for each step, being: 1) hSTARR-seq data with final regions; 2) human EMERGE predicted enhancers; and 3) mouse orthologous RE candidates, based on fetal AVCS-ATAC-seq and mouse EMERGE enhancer prediction, translated to the human genome. The figure now also shows which regions have been included for downstream analysis at each step.

The LacZ-stained heart showing reporter activity driven by Tbx3-eA from Figure 3B has indeed been published previously, and as such we omitted it from the figure. We believe that the message of this panel, to show that this region (and previously identified Tbx3-eB, not shown here) was marked in all datasets used throughout the pipeline, serves as a validation of our approach and as such we kept the genome browser view of this region and moved it to the Figure 3—figure supplement 1. We did not test other RE candidates in vivo.

9) Several of the candidate risk alleles, examined in luciferase activity in Figure 4C and D, show discordant effects on basal activity and TF-stimulated activity. This should be described and considered. Do the variants alter TF binding sites of known activator or repressor TFs that would help explain this discordance? Do any of the analyzed variants alter TF binding sites of the TF analyzed, which may explain their altered regulatory response to TF expression?

We have added a short paragraph to the Discussion on the discordant direction of effect of variant REs on basal activity and SG4- or Tcf-stimulated activity. “Several of the tested risk alleles (e.g. hRE8, hRE9, hRE18) showed a discordant direction of effect between basal activity in COS-7/HL1 and SG4-/Wnt-stimulated activity. This observation suggests that potentially multiple factors act on these variant REs, with SG4- or Tcf factors counteracting the effect of regulatory networks active in COS-7 or HL1 cells. Although we could not identify specific TF binding motifs affected by the respective SNPs in these REs, studying TF occupancy in more detail specifically in AVCS cells could elaborate on the precise mechanisms through which these variant REs act.”

Furthermore, we have analyzed whether the variants in the tested REs disrupt or create transcription factor binding sites using JASPAR. To this end, we took the variant nucleotide and 10 nucleotides both up- and downstream, and used these 21bp sequences as input for transcription factor binding motif recognition in JASPAR using default settings. We did not find any disrupted or de novo created binding motifs this way, although we are aware that the JASPAR binding motif database is incomplete and/or inaccurate as described below (#11). We also manually analyzed the variant REs for disrupted/de novo created binding motifs, e.g. T-box/E-box motifs, but did not find any. We have added this to the Results section and added a paragraph in the Discussion section.

10) In Figure 6, the schematic shows the location of the TALEN cuts. The authors should delineate VR1 and VR2, as in Figure 5.

We edited Figure 6 such that it shows the location of VR1 and VR2, additional to the location of the CRISPR/TALEN deletion sites according to the reviewer’s suggestion.

11) Figure 1C do the authors have insight as to why the Motif consensus for HAND2, TCF3, and TCF4 do not contain a clear E-box motif such as shown for MyoD? The SOX9 Motif also seems out of place with the canonical consensus: CCTTGAG. SMAD 3&4 likewise are reported to bind CAGAC, CAGCC as well as the 5-bp consensus sequence GGC(GC)|(CG). What is shown in the motif is very different and some discussion of this is warranted there is not confidence that the aforementioned transcription factors would bind these sequences robustly. There is simply low confidence that the algorithm is pulling accurate motif data and the authors should try to address this in the text.

We agree with the reviewer, and have specified interpretations of the Motif analysis. The confidence of such analyses depends on the quality of motifs within the database, which are experimentally derived by different experiments, in various cell types or developmental stages. As such, scanning sequences for motif enrichment using HOMER or other large scale analysis tools should be interpreted with this in mind. Nevertheless, we used the analysis to obtain a global impression of the factors that could be involved in AVCS Tbx3 regulation, while being aware that the output list is far from comprehensive. We have toned down statements regarding the conclusion of this analysis in the Results and Discussion section, and have specified in the Discussion that we used this analysis mainly to obtain an impression of involved factors, and have discussed the caveats of this analysis. Furthermore, we have removed the motif logos from Figure 1C.

12) Figure 2 same issue with consensus data in 2G (note the motif name and the motif do not line up correctly).

Similar to the response to the question above, we have removed the motif logos from Figure 2G and corrected the misalignment of motif name and p-value. Furthermore, similarly to the clarification to the previous question, we have discussed in the manuscript that the motif analysis here was done primarily to see if the regulatory activity of the identified STARR-seq regions indeed correlated with enrichment of motifs for the respective co-transfected factors, rather than to identify novel factors.

13) Figure 3 some clarity in the narration would be helpful. Paragraph one of subsection “Identification of variant REs within VR1 and VR2” discuss 9 candidate REs in VR1 and 13 in VR2. Are these regions supposed to be identifiable in Figure 3A (as is the assumption)? The figure shows 6 candidate REs which resolve to 3 with SNP overlaps comparing the figure to the narration is confusing.

We agree with the reviewer that the narration in this figure might be unclear. Figure 3 served as simplified example to illustrate the workflow we used to identify regulatory elements within VR1 and VR2. As such, the regions shown in Figure 3 do not depict the true REs we identified in this manuscript. We have revised this figure and used the actual data to illustrate the workflow, simultaneously showing which regions have been included for downstream analysis at each step.

14) Figure 6C shows only Tbx3 expression Tbx5 and Med13l expression narrated as being in Figure 6C. Panel G would benefit from slightly enlarged images.

We have corrected the text to include the notion that expression levels of Tbx5 and Med13l have been measured but are not shown in Figure 6C. Furthermore, we have enlarged panel G for clarity, following the reviewer’s suggestion.

15) The criteria and datasets used to define the EMERGE regions is not found in this paper nor is there any citation listing where these regions came from. Please clarify.

We have added a paragraph in the Materials and methods section describing the use of EMERGE for the prediction of regulatory elements in both the mouse and human genome. Furthermore, we have added Supplementary Tables 10 and 11 to Supplementary file 1, listing the datasets used by EMERGE to generate a genome-wide prediction track of cardiac enhancers in both the mouse and human genome.

16) What was the criteria used to identify the 9 and 13 (as listed in the text) candidate regulatory elements in VR1/2? Supplementary Table 4 lists only 20 candidate regulatory elements (7 and 13). What happened to hRE1 and hRE2 to exclude them from downstream analysis and inclusion in the table? Were other sites excluded as well? It is also unclear from Figures 2H-K and Figure 3A, and Supplementary Tables 3 and 4 what was used to define the candidate regulatory elements examined in greater detail in Figures 4, Figure 3—figure supplement 1 and S3. For example, hRE13 was not examined in Figures 4C/D or Figure 4—figure supplement 1. Overall, I really like the multi-factor approach to defining candidates; however, the key filtering steps taken to get the final list of candidates needs to be more clearly defined.

We have edited Figure 3, and used the actual *TBX3* locus and data to illustrate the pipeline and the filtering steps that were used to identify candidate REs. Potential regulatory elements have been identified throughout the entire human *TBX3* locus, based on the three approaches/filtering steps shown in revised Figure 3: 1) human STARR-seq activity, 2) human EMERGE enhancer prediction, and 3) mouse orthologous RE candidate regions, based on fetal AVCS-ATAC-seq and mouse EMERGE, translated to the human genome. We then looked at which of these regions were located within VR1 or VR2, and included these regions in Supplementary file 1—Supplementary Table 4. These regions were then overlapped with the GWAS SNPs for PR interval and QRS duration, and the REs overlapping one or more of these SNPs were used for variant haplotype activity testing using transfection/luciferase assays. We have revised the text accordingly to correspond to the revised figure, and hope that this revision clarifies the filtering steps we used to obtain the final list of RE candidates.

hRE1 and hRE2 were initially included as potential regulatory elements. However, their genomic location was outside the set boundaries of VR1 for final downstream analysis and inclusion – we have corrected the manuscript accordingly, stating that 7 and 13 REs have been identified in VR1 and VR2, respectively.

Due to technical issues we failed to amplify and clone hRE13 from the genomes of the human haplotypes we used. We have added this notion to the manuscript.